# Electroactive materials with tunable response based on block copolymer self-assembly

Ivan Terzic[1], Niels L. Meereboer[1], Mónica Acuautla[2], Giuseppe Portale [1] & Katja Loos [1]

Ferroelectric polymers represent one of the key building blocks for the preparation of flexible electronic devices. However, their lack of functionality and ability to simply tune their ferroelectric response significantly diminishes the number of fields in which they can be applied. Here we report an effective way to introduce functionality in the structure of ferroelectric polymers while preserving ferroelectricity and to further tune the ferroelectric response by incorporating functional insulating polymer chains at the chain ends of ferro-electric polymer in the form of block copolymers. The block copolymer self-assembly into lamellar nanodomains allows confined crystallization of the ferroelectric polymer without hindering the crystallinity or chain conformation. The simple adjustment of block polarity leads to a significantly different switching behavior, from ferroelectric to antiferroelectric-like and linear dielectric. Given the simplicity and wide flexibility in designing molecular structure of incorporated blocks, this approach shows the vast potential for application in numerous fields.

[1] Macromolecular Chemistry and New Polymeric Materials, Zernike Institute for Advanced Materials, University of Groningen, Nijenborgh 4, 9747AG Groningen, The Netherlands. [2] Nanostructures of Functional Oxides, Zernike Institute for Advanced Materials, University of Groningen, Nijenborgh 4, 9747AG Groningen, The Netherlands. These authors contributed equally: Ivan Terzic, Niels L. Meereboer. Correspondence and requests for materials should be addressed to K.L. (email: k.u.loos@rug.nl)

Nowadays, ferroelectric polymers, together with semi-conducting polymers, portray as essential elements for future flexible organic devices due to their light weight, flexibility and processability[1]. Poly(vinylidene fluoride) (PVDF) and its copolymer with trifluoroethylene (P(VDF-TrFE)), which exhibit a large polarization, high dielectric constant, enviable thermal and chemical stability, are the most used ferroelectric polymers[2,3]. The C-F bond orientation inside the highly polar β-crystalline phase with all dipoles aligned perpendicular to the main chain allows their switching under an applied electric field, while the high packing density of crystals prevents dipole dis-orientation after field removal, leading to the desirable hysteresis behavior[4,5].

Introduction of extra functionalities into the structure of fer-roelectric polymers can further widen their application in related fields[6–10]. The response of PVDF-based polymers to an applied electric field can be altered from ferroelectric to relaxor or even linear dielectric by the adjustment of the crystalline domain size or by screening the nominal electric field values inside the crys-talline ferroelectric phase[11–17]. Thus, it would be highly beneficial to develop a simple method that can simultaneously tailor the hysteresis shape while adding a new functionality. Self-assembly processes are one of the possibilities in this respect, as they emerge as one of the most effective means to construct functional nanomaterials that combine properties of all constituent com-ponents, readily tailored by either changing molecular char-acteristics of a material or under external stimuli[18–20]. In addition to the creation of functional nanostructures, self-assembly brings in the possibility to control and adjust the structure and prop-erties of bulk materials and to induce anisotropy at multiple length scales[21–23]. In polymers, one of the simplest ways to accomplish improved tunable functionalities is by the preparation of block copolymers[24,25]. The self-assembly of block copolymers is governed by the balance between the interaction enthalpy and entropic elasticity between immiscible chemically connected blocks. By adjusting the block length, the ratio between the blocks and the temperature, it is possible to obtain lamellar, cylindrical, spherical or gyroid structures in the range between 10 and 100 nm. An even richer diversity of structures can be achieved by simply increasing the number of blocks[26,27], by manufacturing different block architectures[28] or by incorporating small func-tional molecules within the block copolymer structure[29–31].

Current research reports demonstrate that the addition of an insulating functional component into the structure of the PVDF can significantly modify the coupling forces between ferroelectric dipoles and form a good electric shield that can alter the dipole switching behavior[11,32,33]. Nevertheless, research thus far has focused on studying weakly segregated graft copolymer systems in which it is impossible to exclude the negative influence of grafting chains on the ferroelectric phase formation and dipole switching. The defects formed via the growth of amorphous polymer chains hinder the alignment of the dipoles, resulting in antiferroelectric-like behavior already at a low grafting density[11]. It is to expect that the formation of well-separated block copo-lymers has less influence on the crystallization and that ferro-electric properties can be largely preserved compared to graft copolymers. However, due to the absence of synthetic procedures that grant the successful preparation of high molecular weight block copolymers that demonstrate strong phase separation, the ferroelectric properties of PVDF-based block copolymers have not yet been investigated.

Herein, we present an appealing approach to tune the ferro-electric response of the P(VDF-TrFE), based on the covalently linking of functional insulating chains to the chain ends in the form of A-B-A triblock copolymers (Fig. 1a) to provide insight into the factors that affect the shape of hysteresis loops and that can be used for the fine-tuning of the ferroelectric response of the material.

## Results

**Preparation of block copolymers.** The synthetic route applied for the preparation of PVDF-based block copolymers based on the use of functionalized benzoyl peroxide as the initiator of the poly-merization is depicted on Fig. 1b[34–37]. A successful preparation of chlorine-terminated P(VDF-TrFE) is confirmed using [1]H NMR, where end group signals of phenyl protons at 8.07 and 7.65 p.p.m. and methylene protons next to the chlorine atom at 4.80 p.p.m. are detected. The chlorine atoms are fully substituted with azide groups after stirring the polymer with sodium azide in DMF overnight, which is verified with a shift of methylene protons signal from 4.85 to 4.60 p.p.m. (Supplementary Fig. 2).

The azide terminated telechelic P(VDF-TrFE) is subsequently used for the preparation of the block copolymers using copper(I)-catalyzed azide-alkyne cyclo-addition reaction with alkyne terminated P2VP or PS made by reversible addition-fragmentation chain transfer (RAFT)[38–42]. After the reaction is completed, the signal of the methylene protons next to the azide group of (PVDF-TrFE) is fully relocated from 4.60 to 5.80 p.p.m. underneath the peak of TrFE units, verifying the full conversion of P(VDF-TrFE) end groups and successful preparation of the block copolymers (Fig. 2a, b). Figure 2c shows the gel permeation chromatography (GPC) traces of P(VDF-TrFE), PS, P2VP and their block copolymers. It is evident that after the reaction, a negative peak of P(VDF-TrFE) and the peaks of P2VP and PS are no longer detectable. Instead, new signals corresponding to the block copolymer are observed at lower retention volumes, demonstrating the successful synthesis of pure block copolymers. The molecular characteristics of the synthesized polymers are depicted in Table 1.

**Structural characteristics of block copolymers.** Block copolymer films are prepared by solvent casting the samples from DMF (1.0 wt%) at 45 °C and subsequent thermal annealing at 170 °C for 5 min in order to reach the equilibrium structure. The films are then cooled down to induce crystallization and to obtain ~20 μm thick films. Figure 3 shows the small-angle X-ray scat-tering (SAXS) profiles of the block copolymers in the melt at 170 °C and at room temperature after crystallization together with the transmission electron microscopy (TEM) images recorded after crystallization[43,44]. Based on an almost symmetrical composition (with about 60 vol.% of P(VDF-TrFE)), a lamellar morphology of the phase separated melt is expected[45]. Indeed, the SAXS profiles of both block copolymers with P2VP (VDF:TrFE = 70:30 and VDF:TrFE = 50:50) at 170 °C display three relatively strong sig-nals located at positions $q^{\star}$: $2q^{\star}$: $3q^{\star}$ corresponding to a lamellar morphology. The periodicity of the lamellar morphology can be obtained by the position of the first maximum $q^{\star} = 0.17$ nm$^{-1}$ by using the Bragg's law $L = 2\pi/q^{\star} = 37$ nm. Upon cooling from the phase separated melt, the microphase separation driving force is stronger than crystallization, leading to crystallization inside the lamellar nanodomains. No change in the shape of the SAXS profiles upon crystallization is observed, confirming confined crystallization and the conservation of the melt morphology. However, the size of the lamellar spacing is found to decrease to $L = 34$ nm as a result of the density increase upon crystallization. The non-stained TEM images also show the well-ordered lamellar structure with the dark layers corresponding to the crystalline P (VDF-TrFE) and a domain spacing that corresponds to the length scales obtained by SAXS. Compared to block copolymers with P2VP, the samples prepared with non-polar PS demonstrate a considerably different behavior. The TEM image depicts the

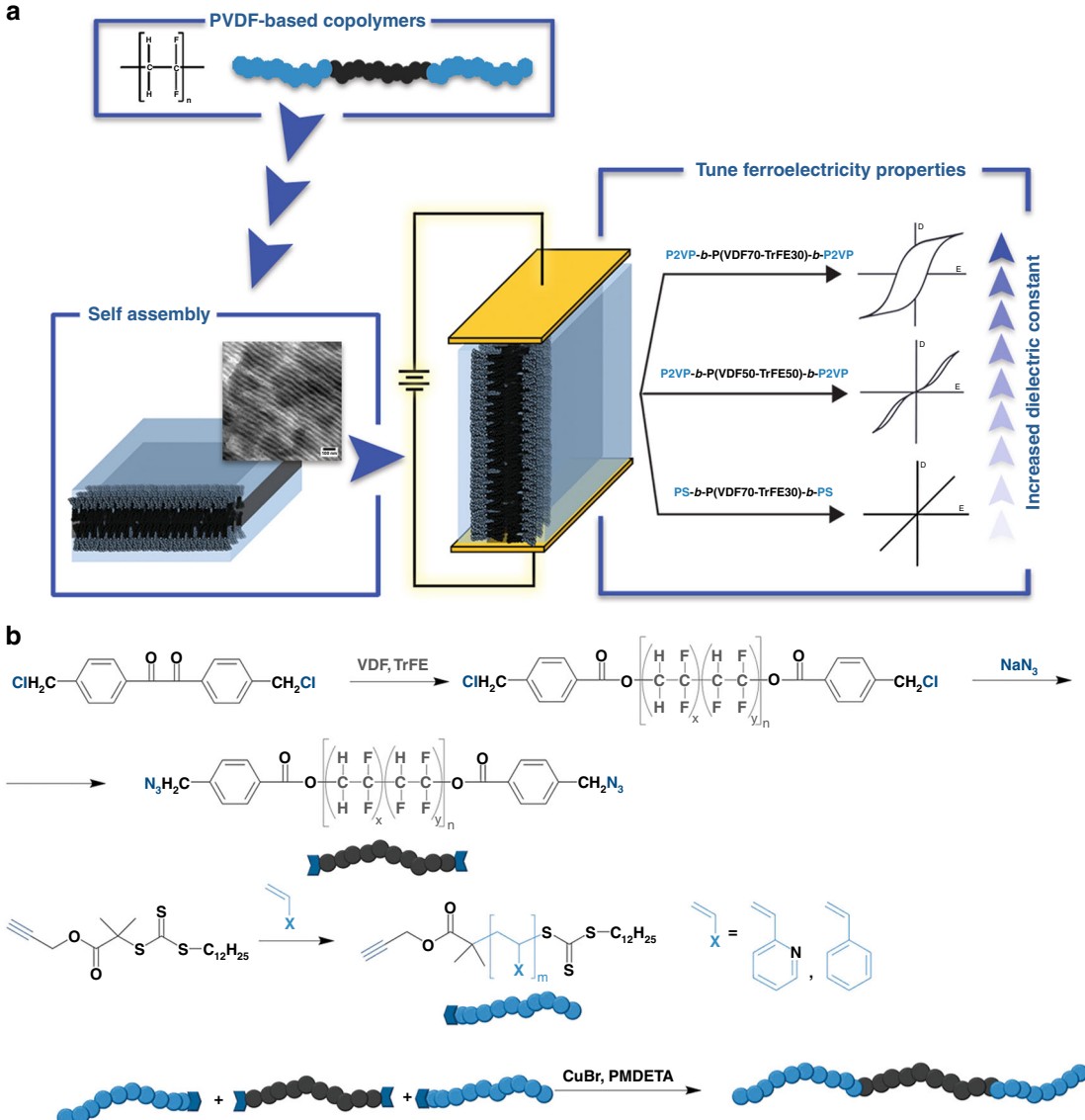

**Fig. 1** Block copolymer approach for the tunable ferroelectric response. **a** Schematic representation of the approach used for tuning of P(VDF-TrFE) ferroelectric properties using block copolymer self-assembly. The response of the block copolymers on the electric field depends strongly on the polarity of both blocks. **b** The synthetic approach applied for the preparation of P(VDF-TrFE) based block copolymers using CuAAc click-coupling of azide terminated P(VDF-TrFE) and alkyne terminated P2VP or PS

formation of P(VDF-TrFE) nanospheres dispersed in the PS matrix without apparent long-range order, as observed in Fig. 3d. This structure is the consequence of a lower χ interaction parameter between P(VDF-TrFE) and PS compared to P2VP. Still, the glass transition temperature ($T_g$) of PS was high enough to prevent the break-out of crystallization and to confine P(VDF-TrFE) segments inside spherical domains.

Differential scanning calorimetry (DSC) allows us to get a better insight into the crystallization mechanism of P(VDF-TrFE) inside block copolymer nanodomains. DSC of pristine P(VDF$_{70}$-TrFE$_{30}$) shows two exothermic peaks (Fig. 3g). The peak at the higher temperature corresponds to the crystallization of the fluorinated blocks into the paraelectric phase ($T_{cr} = 119.5$ °C), while the peak at 52.1 °C is related to a paraelectric-to-ferroelectric Curie transition ($T_c$) (ref. [46] and N.L.M., manuscript submitted). An increase in TrFE amount leads to the disappearance of the Curie transition, and only a crystallization peak at 130 °C is detected[36]. The DSC traces of all block copolymers with a lamellar morphology do not demonstrate a large difference in

shape compared to the neat P(VDF-TrFE) copolymers. Impurities forming crystallization nuclei inside lamellar forming block copolymers are present in every lamella inducing heterogeneous nucleation that is followed by long-range crystal growth. As a result, crystallization starts at low or no undercooling[47,48]. In contrast to this, a reduction of the $T_{cr}$ is observed for the PS-b-P (VDF$_{70}$-TrFE$_{30}$)-b-PS, suggesting strong confined crystallization of the fluorinated blocks inside the spherical domains. The significant degree of undercooling for the crystallization inside isolated spherical domains is a direct consequence of a different crystallization mechanism. In addition to the fact that the crystallization inside the nanospheres is highly frustrated, the number of spherical domains highly exceeds the number of impurities, so a homogeneous nucleation process dictates the crystallization, causing the reduction of $T_{cr}$ from 120 to 80 °C[48,49].

Considering that block copolymers of P(VDF$_{70}$-TrFE$_{30}$) with PS and P2VP show different morphologies in the melt and, therefore, different crystallization nature, the difference in their

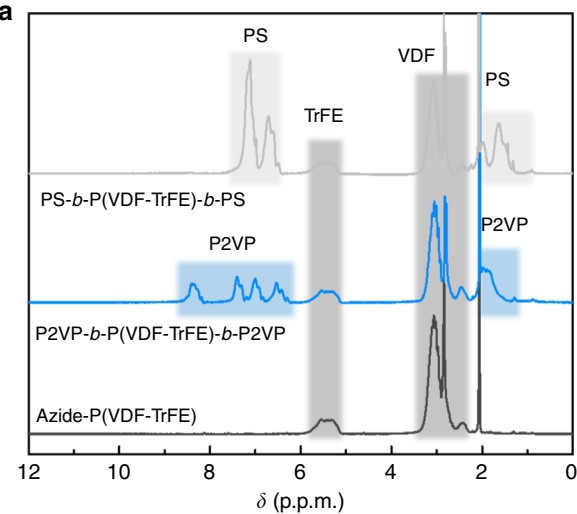

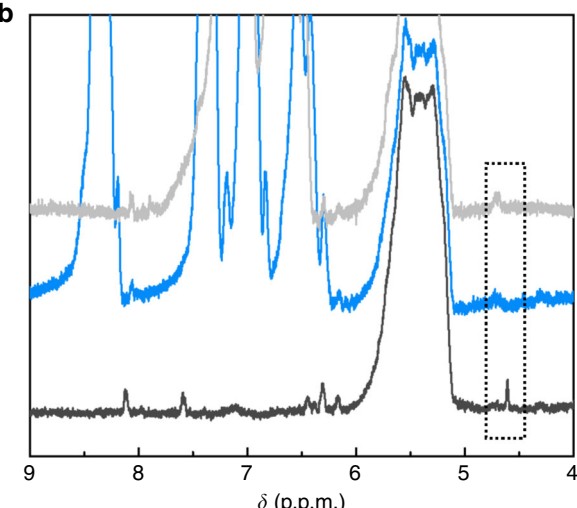

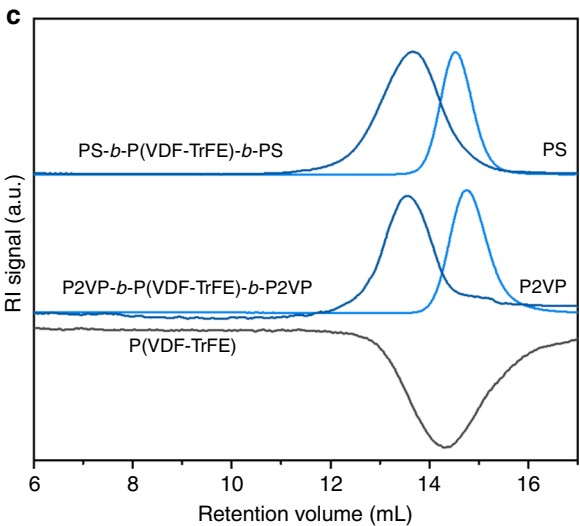

**Fig. 2** Synthesis of block copolymers. **a** [1]H NMR spectra of telechelic P(VDF-TrFE) and corresponding block copolymers with P2VP and PS; P(VDF-TrFE), PS and P2VP peaks are highlighted with dark gray, light gray and blue, respectively. **b** Enlarged [1]H NMR spectra demonstrate complete shift of the methylene protons signal at 4.60 ppm and full conversion of P(VDF-TrFE) azide end groups. **c** GPC of P(VDF-TrFE) and its block copolymers. All elugrams are obtained in THF at a flow rate 1.0 mL min$^{-1}$ at 35 °C. In THF, P(VDF-TrFE) gives negative R. I. signal due to its low d$n$/d$c$ value

lamellar structures are formed with thin crystalline PVDF-TrFE layers (Fig. 3e and Supplementary Fig. 3a), due to the exclusion of structural defects from the crystalline phase formed during casting from DMF. Annealing in the more mobile paraelectric phase allows defects to rearrange and to increase the sample crystallinity[50,51]. Consequently, the growth of crystalline layer is observed (Fig. 3f and Supplementary Fig. 3b) while the high $T_g$ ($T_g \approx 100$ °C) of both PS and P2VP prevents the change of the lamellar periodicity, as proven by temperature resolved SAXS (Supplementary Fig. 4). As a result of the thermal annealing process, symmetrical lamellar morphologies with increased crystallinity are obtained for both block copolymers, allowing the comparison of the ferroelectric properties of BCPs with the same morphology but different chemical composition (i.e. polarity of the non-fluorinated block).

In order to better understand the influence of the incorporation of the amorphous polymer on the crystalline behavior of block copolymers, the crystalline structure was investigated by wide-angle X-ray scattering (WAXS) (Fig. 4). The pristine P(VDF$_{70}$-TrFE$_{30}$) exhibits a so called low temperature ferroelectric phase (LTFE) with the (110/200)$_{LTFE}$ reflection located at 14.1 nm$^{-1}$, characteristic for the all-trans crystal conformation[52]. A similar shape of the WAXS profile and the same crystalline phase are observed for both block copolymers (with P2VP and PS) after annealing in the paraelectric phase. However, the inclusion of extra TrFE units inside the P (VDF-TrFE) backbone induces changes in the crystalline nature of the polymer. The deconvolution of the crystalline peak from P2VP-$b$-P(VDF$_{50}$-TrFE$_{50}$)-$b$-P2VP reveals two crystalline phases present inside the sample. The reflection located at 13.5 nm$^{-1}$ corresponds to the cooled ferroelectric phase (CLFE) that consists mostly of trans sequences with some gauche defects[50,52]. It is important to note that the (110/200) $d$-spacing, which strongly influences the dipole switching mechanism, displays an increase from 0.445 nm to 0.465 nm after the incorporation of more TrFE units. The second crystalline peak located at 13.1 nm$^{-1}$ is related to the high temperature paraelectric phase (HTPE) with $d$-spacing of 0.485 nm. These findings are in agreement with the results obtained by Lovinger et al. for TrFE rich P(VDF-TrFE) copolymers[53]. Ferroelectric properties of a material are primarily related to its overall crystallinity and, therefore, these values are calculated from WAXS ignoring the dilution effect of the amorphous component summarized in Table 1. As expected, the crystallinity of block copolymers is reduced compared to the pristine P(VDF-TrFE) due to the inclusion of ca. 30 wt% of non-crystalline P2VP and PS into the calculation. After considering the dilution effect of the amorphous blocks, a negligible reduction in crystallinity is observed for all block copolymers. Such a minor reduction of crystallinity compared to the pristine polymer is a consequence of strong phase separation between blocks. Therefore, the higher miscibility between the PS and P(VDF$_{70}$-TrFE$_{30}$) blocks (which again reflects a lower $\chi$ parameter) is most probably the cause of the reduced crystallinity compared to block copolymer with P2VP. It is worth noting that the crystallinities obtained from WAXS profiles generally matched those determined by DSC (see Table 1) except

switching behavior will not simply be a consequence of the difference in polarity. Fortunately, the lamellar morphology of both block copolymers is obtained via solvent casting from DMF and subsequent thermal annealing at 120 °C between the Curie and the melting temperature. During solvent casting asymmetric

**Table 1 Molecular characterization data for P(VDF-TrFE) and its block copolymers**

| Entry | Molecular weight (g mol$^{-1}$) | Đ | $f_{P(VDF-TrFE)}$[a] (wt.%) | $T_{cr}$ (°C) | $T_c$ (°C) | $X_c$[b] (%) | $X_c$[c] (%) | $X_c$[d] (%) |
|---|---|---|---|---|---|---|---|---|
| P(VDF$_{70}$-TrFE$_{30}$) | 28,040[e] | 1.45 | 100 | 119.5 | 52.1 | 38 | 38 | 39.5 |
| P2VP-b-P(VDF$_{70}$-TrFE$_{30}$)-b-P2VP | 32,850[f] | 1.80 | 70 | 119.2 | 51.0 | 22 | 36.5 | 38 |
| PS-b-P(VDF$_{70}$-TrFE$_{30}$)-b-PS | 35,940[f] | 1.95 | 65 | 80.0 | 54.5 | 18 | 34 | 32.5 |
| P2VP-b-P(VDF$_{50}$-TrFE$_{50}$)-b-P2VP | 34,280[f] | 1.72 | 70 | 127.0 | n.a. | 37 | 52.5 | 44 |

[a]Weight fraction of P(VDF-TrFE) determined using $^1$H NMR
[b]Overall crystallinity($X_c$) calculated from WAXS
[c]True crystallinity values after normalization to the P(VDF-TrFE) volume percentage
[d]Degree of crystallinity calculated form DSC using the following equation: $X_c = (\Delta H_c / \Delta H_{100}) \times 100\%$. $\Delta H_c$ was determined based on DSC thermograms and normalized to the P(VDF-TrFE) weight percentage. $\Delta H_{100} = 42$ J g$^{-1}$ for crystallization in the paraelectric phase
[e]Determined using GPC
[f]Molecular weight calculated from $M_{n,GPC}$ values of P(VDF-TrFE) taking in the account ratio between the blocks using $^1$H NMR (Equations in Supplementary Note 3), the molecular weight of P(VDF-TrFE) used for the synthesis of block copolymers was 22,500 g mol$^{-1}$

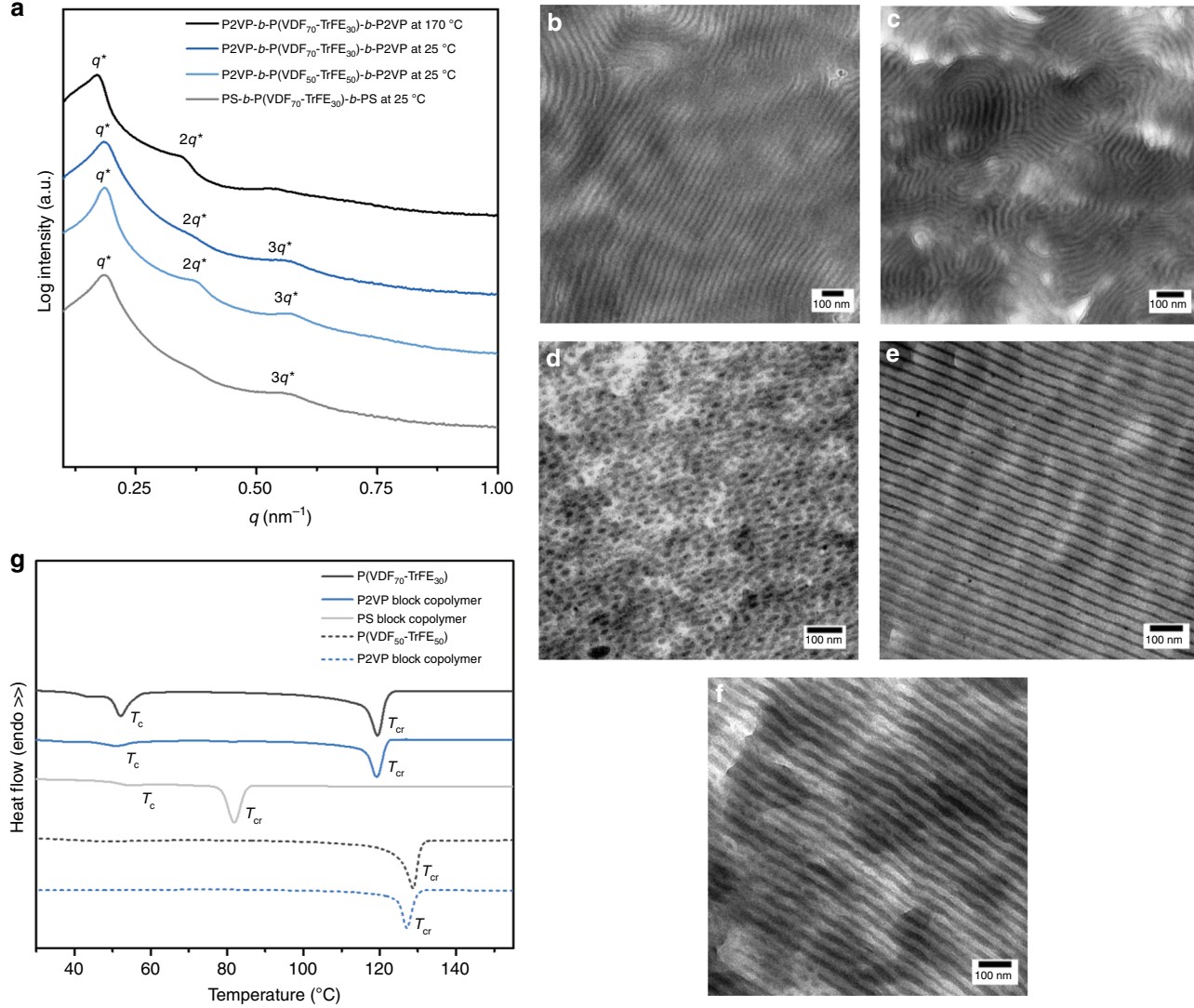

**Fig. 3** Structural characteristics of block copolymers. **a** SAXS profiles of the block copolymers prove the confinement of the crystallization inside the nanodomains formed in the melt state. TEM images of **b** P2VP-b-P(VDF$_{70}$-TrFE$_{30}$)-b-P2VP, **c** P2VP-b-P(VDF$_{50}$-TrFE$_{50}$)-b-P2VP and **d** PS-b-P(VDF$_{70}$-TrFE$_{30}$)-b-PS after crystallization from the melt demonstrate different segregation strength between blocks. No staining of the block copolymers is required as sufficient density contrast exists between crystalline P(VDF-TrFE) and amorphous blocks. Annealing of **e** the solvent casted PS-b-P(VDF$_{70}$-TrFE$_{30}$)-b-PS with asymmetric lamellar morphology at 120 °C results in **f** the increase of the crystalline layer thickness without changing the overall lamellar period (See Supplementary Fig. 4). **g** DSC cooling curve of the pristine P(VDF-TrFE) and corresponding block copolymers, obtained at a cooling rate 10 °C min$^{-1}$

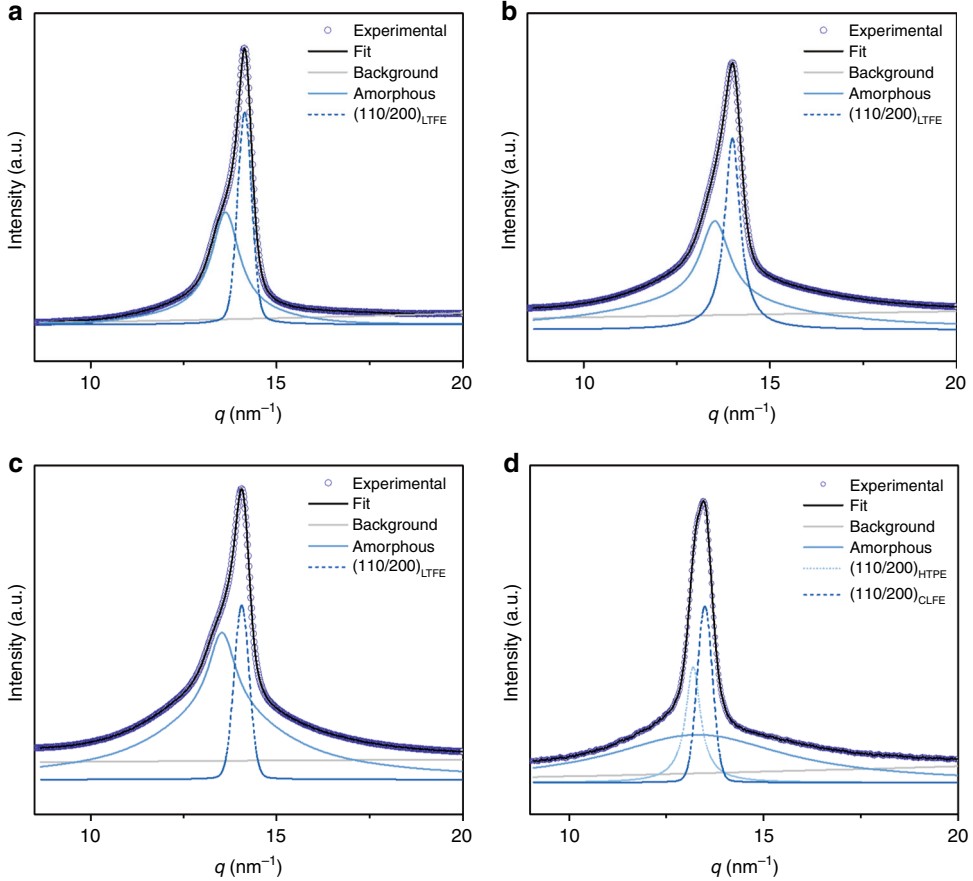

**Fig. 4** Crystalline phase of P(VDF-TrFE). WAXS profiles of **a** P(VDF$_{70}$-TrFE$_{30}$), **b** P2VP-*b*-P(VDF$_{70}$-TrFE$_{30}$)-*b*-P2VP, **c** PS-*b*-P(VDF$_{70}$-TrFE$_{30}$)-*b*-PS, **d** P2VP-*b*-P(VDF$_{50}$-TrFE$_{50}$)-*b*-P2VP. Peak fitting is performed to determine the crystalline phases and overall crystallinity of the polymer samples. The experimental profiles were deconvoluted by using the sum of a linear background, and few pseudo-Voigt peaks describing the scattering from the amorphous and the different crystalline phases. For WAXS profile of pristine P(VDF$_{50}$-TrFE$_{50}$) see Supplementary Fig. 5

in the case of P2VP-*b*-P(VDF$_{50}$-TrFE$_{50}$)-*b*-P2VP in which lower crystallinity is obtained after integrating the DSC thermogram. The reason of this mismatch is that the $\Delta H_{100}$ used for the calculations of crystallinity are dependent on the TrFE content and are therefore not known exactly. This causes the deviation from the values obtained using WAXS, that can be considered as the more accurate method.

**Dipole switching characteristics of block copolymer.** Ferroelectric properties of the P(VDF-TrFE) and its block copolymers are studied by D–E loop measurements using a bipolar triangular wave form at a frequency of 10 Hz (Fig. 5a). Figure 5b depicts the bipolar D–E loops for P(VDF$_{70}$-TrFE$_{30}$). Upon applying an electric field, chain rotation and alignment of the dipoles occurs in the direction of the field. However, the dipole disorientation rate is reduced due to a high packing density of PVDF crystals and a strong coupling force between neighboring domains, leading to a broad hysteresis loop. The alignment of the crystalline dipoles induces a local polarization $P_{in}$ that is compensated with the polarization at the crystalline-amorphous interface-compensational polarization ($P_{comp}$)[11,54]. While the charge separation at the electrodes, together with the compensational polarization, creates a polarization field ($E_{pol}$), the $P_{in}$ results in the formation of a depolarization field ($E_{dep}$) in the opposite direction. The dipole reversal process and the shape of the ferroelectric loop are a direct consequence of the relationship between these two local fields[54]. In the case P(VDF$_{70}$-TrFE$_{30}$),

$E_{dep}$ turns out to be always lower than $E_{pol}$ above the coercive field, resulting in rectangular shaped ferroelectric loops.

The incorporation of P2VP chains at both ends of P(VDF$_{70}$-TrFE$_{30}$) and their phase separation do not induce drastic changes in the shape of the D–E loops, as shown in Fig. 5c. However, a slightly higher coercive field and lower polarization compared to the pristine P(VDF-TrFE) are observed. Importantly, the same switching behavior is obtained for samples prepared via thermal annealing in the melt and in the paraelectric phase (Supplementary Fig. 6). P2VP has medium polarizability ($\varepsilon_{r,P2VP}$ = 5.5 at 10 Hz)[55], lower than that of the amorphous P(VDF-TrFE). Consequently, compensational polarization at the crystalline-amorphous interface is reduced, resulting in a decrease in the electric field $E_{pol}$. However, due to the small reduction of the dielectric constant of the amorphous phase, $E_{pol}$ is still higher than the depolarization field $E_{dep}$, at all applied fields, which preserves the ferroelectric properties.

The difference in the dielectric constant between two lamellar layers gives rise to an uneven distribution of the electric field inside them[56]. The nominal field in the crystalline layer is lower than the applied external electric field. Thus, higher fields compared to the pure P(VDF-TrFE) have to be applied in order to achieve dipole flipping. Nevertheless, the formation of the crystalline-amorphous layered structure increases the distance between ferroelectric crystalline domains, while the number of the adjacent domains is reduced. This generates weakened coupling and easier switching between ferroelectric domains. Consequently, the opposite influence of both factors on dipole

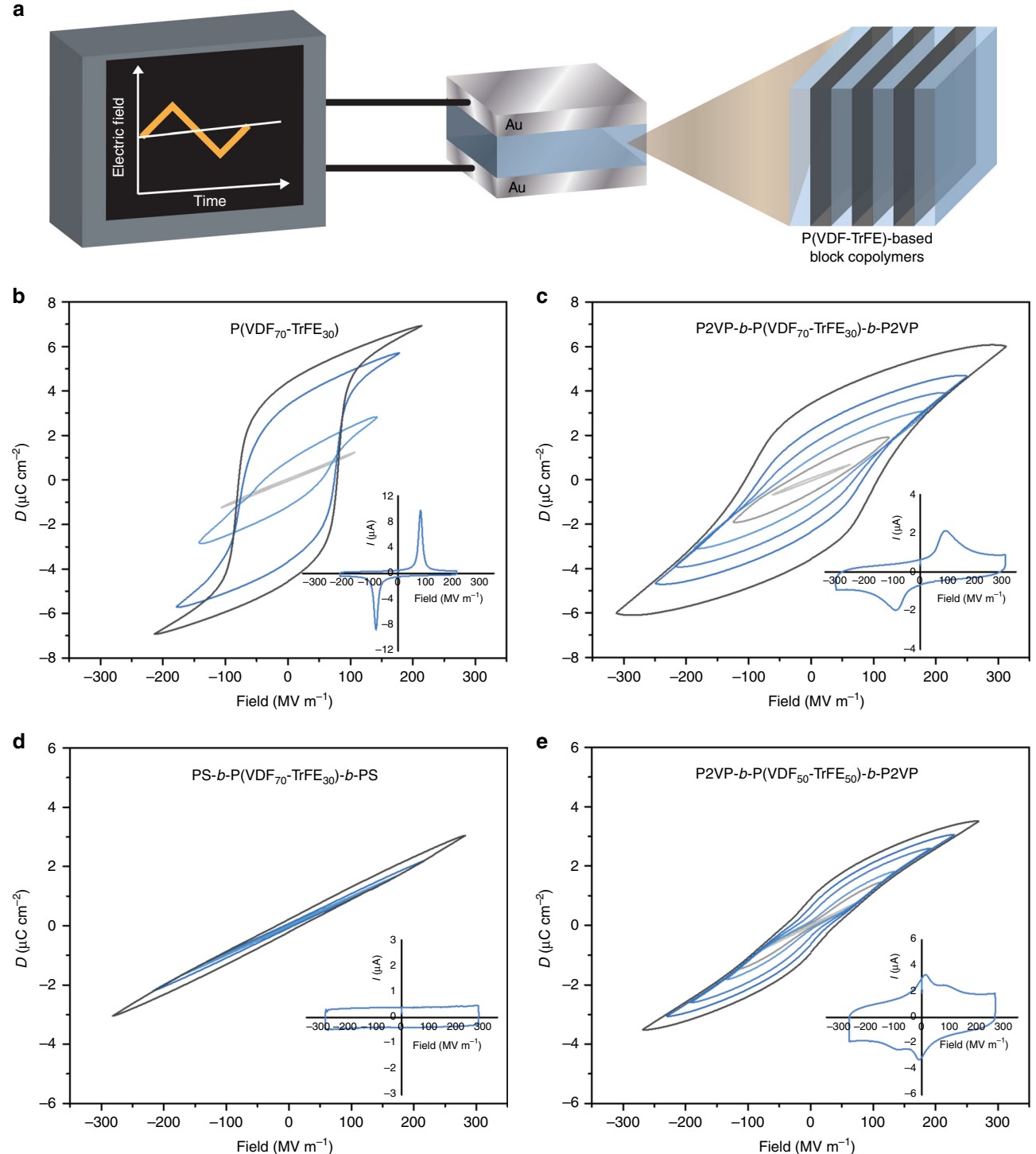

**Fig. 5** Response of block copolymers to the applied electric field. **a** Schematic representation of the measurement setup and devices used for the measurement of the hysteresis loop shape. An AC voltage is applied over a polymer sample sandwiched between gold electrodes. The obtained bipolar $D$–$E$ hysteresis loops for **b** P(VDF$_{70}$-TrFE$_{30}$), **c** P2VP-$b$-P(VDF$_{70}$-TrFE$_{30}$)-$b$-P2VP, **d** PS-$b$-P(VDF$_{70}$-TrFE$_{30}$)-$b$-PS, **e** P2VP-$b$-P(VDF$_{50}$-TrFE$_{50}$)-$b$-P2VP, obtained at different applied electric fields until electric breakdown. For the better understanding of the switching mechanism, $I$–$E$ curves are depicted in the inset. The switching characteristics of pristine P(VDF$_{50}$-TrFE$_{50}$) are described elsewhere[64]. Note that all block copolymers demonstrated higher breakdown strength compared to the pristine P(VDF-TrFE)

switching creates only a slight increase of the coercive field. Additionally, less coupled ferroelectric domains and high amount of dielectric P2VP (30 wt.%) are the main reasons for the reduced polarization in comparison to the parent P(VDF-TrFE).

Figure 5e reveals the $D$–$E$ loops of P2VP-$b$-P(VDF$_{50}$-TrFE$_{50}$)-$b$-P2VP, characterized with double hysteresis loops and low remanent

polarization typical for antiferroelectric materials. In order to better understand the switching mechanism inside this material, we examined the shape of the current-electric field ($I$–$E$) curve (inset of the Fig. 5e). Using the $I$–$E$ measurement, every polarization reversal step is visualized, providing us with extra information about the mechanism. Instead of only one switching event, as for P2VP-$b$-P

(VDF$_{70}$-TrFE$_{30}$)-$b$-P2VP, two peaks in the $I–E$ curve can be distinguished. Providing evidence that the presence of P2VP lamellae does not impede dipole alignment and disorientation, antiferroelectric-like nature of this block copolymer is a result of a different crystalline structure of P(VDF$_{50}$-TrFE$_{50}$) (Supplementary Fig. 5), as well as reduced dipole moment per chain (dipole moment of TrFE (1.4 D) is 1.5 times lower than of β-PVDF units). As explained before on the basis of WAXS results, two coexisting crystalline phases, cooled ferroelectric and paraelectric, are found in this block copolymer. The (110/200) $d$-spacing of both phases is higher than the spacing between ferroelectric all-trans crystals. As a consequence, the crystal packing density is lowered allowing easier dipole and chain flipping. Therefore, the peak on the $I–E$ curve appearing at a lower electric field (14.5 MV m$^{-1}$) corresponds to the fast alignment of dipoles from the paraelectric crystalline phase. The second peak on the $I–E$ curve is closely related to the dipole switching inside the cooled ferroelectric phase. Additionally, during forward poling at high enough fields $E_{pol}$ is higher than $E_{dep}$ allowing dipoles to orient along the field direction. Upon reverse poling, the reduction of $P_{comp}$, caused by the lower dipole moment per chain per repeat unit compared to PVDF$_{70}$-TrFE$_{30}$ leads to a scenario in which $E_{dep}$ becomes higher than $E_{pol}$ at low fields. Hence, the removal of the electric field is accompanied by dipole disorientation, which induces propeller shape antiferroelectric-like behavior. The maximum polarization of this sample is slightly reduced compared to the ferroelectric block copolymer, due to the higher amount of lower dipole moment TrFE units.

An exchange of P2VP with PS of the same molecular weight generates considerably different switching characteristics (Fig. 5d). $D–E$ loops become narrower, resembling linear dielectric behavior, with almost zero remanent polarization and the maximum polarization lower than for P2VP-$b$-P(VDF$_{70}$-TrFE$_{30}$)-$b$-P2VP. Moreover, no peaks corresponding to dipole flipping are detected on $I–E$ curves. PS and P2VP have the same $T_g$ and similar elastic modulus values that are proven to influence the rate of dipole reversal[32]. The main difference that can affect the switching process is their polarity ($\varepsilon_{r,PS} = 2.5$ vs $\varepsilon_{r,P2VP} = 5.5$ at 10 Hz)[55], caused by the replacement of carbon atom with a nitrogen in the aromatic side ring. As already mentioned, the $P_{comp}$ is a function of the number of polarizable dipoles and their polarizability. Therefore, $P_{comp}$ is reduced in the block copolymer with non-polar PS and, as a consequence, $E_{pol}$ is lowered. It is proven already that $E_{dep}$ has its origin in the crystalline phase and is not influenced by the chemical structure of the amorphous part of the material[11]. Particularly for this block copolymer, the values of $E_{dep}$ are above the values of $E_{pol}$ at all measured electric fields, causing no switching of dipoles along the direction of the field. In fact, dipoles are only allowed to wiggle locally giving low values of the maximum polarization. The same behavior and the absence of ferroelectric switching have been already observed for P(VDF-TrFE-CTFE)-$g$-PS graft copolymers and P(VDF-TrFE)/polycarbonate nanolayer films, proving that the same type of nanoconfinement can be induced by block copolymers with a non-polar block[12,33].

## Discussion

We designed a simple method for the preparation of ferroelectric polymers with improved and tunable properties using a block copolymer approach. In this work, P(VDF-TrFE)-based block copolymers are synthesized and their dipole switching behavior is elucidated. The results demonstrate that the main parameter that affects the switching nature of block copolymers is the polarity of the amorphous block. The choice of the block strongly influences the value of the compensational polarization at the amorphous-crystalline interface, responsible for the dipole reversal. The use of a polar P2VP block that phase separates from P(VDF-TrFE) is

critical for the preservation of ferroelectricity inside block copolymers. Conversely, linear dielectric properties with narrow hysteresis loops and almost zero remanent polarization were demonstrated after the incorporation of non-polar PS blocks. Due to the low polarizability of the PS interfacial layer, a reduction of the compensational polarization was caused, as well as a decrease of the polarization field below the values of the depolarization field. Additionally, when more TrFE units (≥50 mol%) are included in P2VP-$b$-P(VDF-TrFE)-$b$-P2VP, instead of only the ferroelectric phase, a mixture of paraelectric and ferroelectric phase is obtained, resulting in an antiferroelectric-like behavior. The incorporation of the functional insulating block does not only grant the tunable response of the ferroelectric polymer, but can potentially deliver additional benefits to the material, such as improved dispersion of nanoobjects[57,58] or any other functional component using supramolecular approaches[59], the preparation of nanoporous ferroelectric materials after selective removal of amorphous block[60], better adhesion to the electrodes[6,7], reduced conducting and dielectric losses[33] and better film formation. Although still exemplified on the proof-of-concept materials, these findings pave the way for developing improved functional materials for advanced applications by using linear ferroelectric block-copolymers[61,62].

## Methods

**Synthesis of block copolymers**. 300 mL of 4-(chloromethyl)benzoyl peroxide (0.1 g, 0.3 mmol) solution in an anhydrous acetonitrile was introduced into a Parr (model 4568) high pressure reactor and purged with N$_2$ to completely remove oxygen from the system. Subsequently, 4 bar of TrFE (6 bar for P(VDF$_{50}$-TrFE$_{50}$)) and 15 bar of VDF were transferred in the reactor, followed by an increase of the temperature to 90 °C. The reaction was allowed to proceed for 4 h under constant stirring. The reaction was stopped by fast cooling to room temperature and depressurization of the reaction mixture to remove unreacted monomers. The solvent was removed in vacuo and the obtained solid was precipitated form DMF in MeOH:water (1:1) and washed twice with methanol and multiple times with dichloromethane to remove the initiator residues. The polymer was finally dried in vacuo at 45 °C to obtain white product. The molar ratio between VDF and TrFE was determined using $^1$H NMR spectra from Equations in Supplementary Note 1. Pristine P(VDF$_{70}$-TrFE$_{30}$) used for ferroelectric measurement was synthesized using 0.05 g of the initiator. $^1$H NMR (400 MHz, acetone-$d6$): δ 8.07 (d, –ArH), 7.65 (d, –ArH), 6.10–5.12 (m, –CF$_2$CHF–), 4.80 (s, –PhCH$_2$Cl), 4.68 (m, –COOCH$_2$CF$_2$–), 3.10–2.70 (m, –CF$_2$CH$_2$–CF$_2$CH$_2$–, head-to-tail), 2.40–2.20 (m, –CF$_2$CH$_2$–CH$_2$CF$_2$–, tail-to-tail).

Chlorine-terminated P(VDF-TrFE) and 10 mol equiv. of NaN$_3$ compared to the end groups were dissolved in DMF and stirred overnight at 60 °C. The polymer solution was concentrated and precipitated three times in MeOH:water (1:1). Subsequent drying of the light-yellow polymer in vacuo at 45 °C yielded azide terminated P(VDF-TrFE). $^1$H NMR (400 MHz, acetone-$d6$): δ 8.07 (d, –ArH), 7.65 (d, –ArH), 6.10–5.12 (m, –CF$_2$CHF–), 4.60 (s, –PhCH$_2$N$_3$), 4.68 (m, –COOCH$_2$CF$_2$–), 3.10–2.70 (m, –CF$_2$CH$_2$–CF$_2$CH$_2$–, head-to-tail), 2.40–2.20 (m, –CF$_2$CH$_2$–CH$_2$CF$_2$–, tail-to-tail).

Alkyne terminated P2VP is prepared as follows. Monomer 2-vinylpyridine (5 mL, 84 mmol), 2-(dodecylthiocarbonothioylthio)–2-methylpropionic acid propargyl ester (RAFT agent) and AIBN (at molar ratio 390:1:0.1) were dissolved in 4 mL of anhydrous DMF and placed in a pre-dried Schlenk tube. The reaction mixture was degassed via at least three freeze-pump-thaw cycles and placed in an oil bath at 70 °C. After 6 h, DMF was removed and the THF solution was precipitated in a large excess of n-hexane. The precipitation procedure was repeated two times to fully remove unreacted species. The obtained light orange powder was dried under vacuum at room temperature for 1 day. $^1$H NMR (400 MHz, CDCl3): δ 8.10–8.55 (m, ArH), 7.22–7.65 (m, ArH), 6.80–7.20 (m, ArH), 6.10–6.75 (m, ArH), 4.83 (m, –S(C = S)S-CH(Ar)–), 4.09 (m, –COO–CH$_2$–), 2.71 (s, –CH, alkyne), 2.4–1.4 (m, –CH$_2$CH(Ar) –), 0.95–0.65 (m, –alkyl). GPC: $M_n = 11500$ g mol$^{-1}$, Đ = 1.21.

Alkyne terminated PS is prepared using following procedure. Styrene monomer (9.6 mL, 84 mmol), RAFT agent and AIBN in a molar ratio 700:1:0.1 were added to a dried Schlenk tube. After three freeze-pump-thaw cycles, the reaction mixture was placed in an oil bath at 70 °C and stirred for the next 10 h. The reaction was terminated by rapid cooling using liquid N$_2$ and the polymer was isolated by precipitation from DMF into a 20-fold excess of methanol. The polymer was collected via filtration and reprecipitated two more times from chloroform by methanol. The resulting polymer was dried overnight in vacuo at room temperature to remove all traces of residual solvent. $^1$H NMR (400 MHz, CDCl3): δ 7.40–6.25 (m, C$_6$H$_5$), 4.83 (m, –S(C = S)–CH(Ar)–), 4.09 (m, –COO–CH$_2$–), 3.27 (–CH$_2$-S(C = S)–), 2.71 (s, –CH, alkyne), 2.40–1.20 (m, –CH$_2$CH(Ar)-), 0.99–0.81 (m, –alkyl). GPC: $M_n = 12800$ g mol$^{-1}$, Đ = 1.20.

A general route for the preparation of P(VDF-TrFE)-based block copolymers is described below. The azide terminated P(VDF-TrFE) (300 mg, 0.016 mmol) and 1.3 equivalents of P2VP or PS compared to end groups of (PVDF-TrFE) were added into dried Schlenk tube. Subsequently, 4 equiv. of copper(I) bromide was introduced and a degassing procedure (three repetitive cycles of evacuating and backfilling with $N_2$) was performed. The polymers and the metal catalyst were dissolved in 4 mL of anhydrous DMF, followed by the addition of PMDETA (30 μl, 0.14 mmol). The reaction was allowed to stir for 3 days in case of PS and 4 days for P2VP at 60 °C, after which it was terminated. The crude reaction mixture was filtered twice using short neutral alumina column in order to remove copper catalyst. The solution was concentrated under reduced pressure and precipitated from THF in a 20-fold excess of hexane for P2VP-b-P(VDF-TrFE)-b-P2VP and MeOH:water (1:1) for the block copolymers containing PS. The light-brown product was collected via filtration and dried overnight in vacuo at room temperature. The unreacted P2VP or PS were removed from the product by washing with a selective solvent. Methanol was used as a selective solvent for P2VP-b-P(VDF-TrFE)-b-P2VP, while diethyl ether showed to be effective for the removal of PS homopolymer. The product collected after purification was dried in vacuo at 45 °C to give pure block copolymers.

**Polymer film preparation**. All the polymers were dissolved in 4 mL DMF (10 mg mL$^{-1}$) and after passing through 0.45 μm PTFE filter casted in an aluminum pan (ø 3 mm). The solvent was allowed to evaporate at 45 °C over two days. Subsequently, the film was heated to 170 °C during 5 min to induce microphase separation. After fast cooling down using air and water lift-off, ca. 20 μm thick free-standing films were obtained. Polymer films annealed at 120 °C were first peeled off and subsequently annealed for 30 min inside the vacuum oven.

**Polymer characterization**. $^1$H Nuclear Magnetic Resonance ($^1$H NMR) spectra were recorded on a 400 MHz Varian (VXR) spectrometer at room temperature. The molecular weight and the dispersity (Đ) of pristine polymers and corresponding block copolymers were determined using triple detection method (refractive index, viscosity and light scattering) using THF, stabilized with BHT, as the eluent at a flow rate 1.0 mL min$^{-1}$ at 35 °C. The separation was carried out by utilizing two PLgel 5 μm MIXED-C, 300 mm columns (Agilent Technologies) calibrated with narrow dispersed polystyrene standards (Agilent Technologies and Polymer Laboratories). Differential Scanning Calorimetry (DSC) thermograms were recorded on a TA Instruments DSC Q1000 by heating the sample to 170 °C, and subsequently cooling down to room temperature at 10 °C min$^{-1}$. Small-angle X-ray scattering (SAXS) and Wide-angle X-ray scattering (WAXS) measurements were carried out at the Dutch-Belgium Beamline (DUBBLE) station BM26B of the European Synchrotron Radiation Facility (ESRF) in Grenoble, France, particularly optimized for polymer investigation[43,44,63].The sample-to-detector distance was ca. 3.5 m for SAXS, ca. 28 cm for WAXS and an X-ray wavelength $\lambda = 0.97$ Å was used. SAXS images were recorded using a Pilatus 1 M detector while WAXS images were recorded using a Pilatus 100KW detector, both with pixel size $172 \times 172$ μm$^2$. The scattering angle scale was calibrated using the known peak position from a standard silver behenate sample. The scattering intensity is reported as a function of the scattering vector $q = 4\pi / \lambda (\sin\theta)$ with $2\theta$ being the scattering angle and $\lambda$ the wavelength of the X-rays. Deconvolution of the WAXS profiles was achieved using MATLAB. The experimental profiles were deconvoluted by using the sum of a linear background, and three to four pseudo-Voigt peaks describing the scattering from the amorphous and the different crystalline phases. Transmission electron microscopy (TEM) was performed on a Philips CM12 transmission electron microscope operating at an accelerating voltage of 120 kV. A piece of the block copolymer film was embedded in epoxy resin (Epofix, Electron Microscopy Sciences) and microtomed using a Leica Ultracut UCT-ultramicrotome in order to prepare ultrathin sections (ca. 80 nm). No additional staining of the samples was performed.

**Hysteresis loop measurements**. The D–E hysteresis measurements were performed using a state-of-the-art ferroelectric-piezoelectric tester aixACCT equipped with a Piezo Sample Holder Unit with a high voltage amplifier (0–10 kV). The AC electric field with a triangular wave form at frequency of 10 Hz was applied over polymer films immersed in silicon oil to avoid arcing and sparking due to the high voltage. The 100 nm thick gold electrodes (ca. 12.5 mm$^2$) with 5 nm chromium adhesion layer were evaporated onto both sides.

## Data availability

The authors declare that the data supporting this study are available within the paper and its Supplementary Information File. All other data is available from the authors upon reasonable request.

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

## Acknowledgements

This work was supported by the Netherlands Organization for Scientific Research (NWO) via a VICI innovational research grant. The authors are very grateful to Prof. Beatriz Noheda for the valuable discussion regarding the ferroelectric measurements, Albert Woortman for the assistance with GPC measurements and Dina Maniar for help preparing images in this manuscript. NWO and the ESRF are acknowledged for granting the beamtime at DUBBLE. Daniel Hermida-Merino is acknowledged for his experimental assistance with the synchrotron experiments.

## Author contributions

The project was conceived and designed by I.T., N.L.M. and K.L. The experimental results were obtained by I.T., N.L.M. and M.A. I.T. and N.L.M. were responsible for the material synthesis, their characterization and device fabrication. M.A. carried out preliminary results on ferroelectric loops measurements and was involved in the discussion of D–E loops results. G.P. was involved in the analysis and discussion about SAXS and WAXS results and deconvolution of the obtained crystalline peaks. I.T. and K.L. wrote the manuscript with useful input from all other authors. All authors discussed results and the paper.

## Additional information

**Competing interests:** The authors declare no competing interests.

