## [Peer Review File · Nature Communications]

Reviewers' comments:

Reviewer #1 (Remarks to the Author):

This manuscript investigated ferroelectric properties of self-assembled PVDF-TrFE ABA triblock copolymers. There are a number of issues. I consider that this manuscript does not meet the high standard of Nat. Commun. Below I list detailed comments.

1. The objective of this work is ill-defined. In the introduction, it states that PVDF (as well as graft copolymers) is not suitable for uniform dispersion of nanoparticles in polymers (i.e., nanocomposites). This is not true! Actually, PVDF is polar and hydrophobic at the same time, and can be much better than simple hydrophobic polymers such as PS or even PMMA for polymer nanocomposites. Regardless of this inaccurate or incorrect statement, this work does not focus on PVDF-based nanocomposites at all. This should be irrelevant.

Next, the Introduction discusses about PVDF-graft copolymers. However, the authors did not fully understand the purpose of those PVDF-graft copolymers. They are intended for electric energy storage (e.g., capacitors), which is the application of linear dielectrics, not ferroelectrics (which is nonlinear dielectrics). Actually, ferroelectricity is UNDESIRE for capacitors, because most of the charged energy cannot be discharged due to energy storage in aligned/poled ferroelectric domains (also see the Discussion section). It is true that PVDF block copolymers are different from PVDF graft copolymers, because the ferroelectric properties can be largely preserved for the PVDF blocks. However, they are simply not useful at all for capacitors!

Third, if these PVDF-TrFE block copolymers are intended for piezoelectric applications, they are not good either, because the dipole density is reduced by the nonferroelectric blocks.

Finally, it is not clear what these PVDF-TrFE block copolymers are good for? What is the exact goal of this work?

2. Synthesis: The click chemistry to make PVDF block copolymers is not very new. A similar (although not exactly the same) approach has already been reported; see: JACS 2015, 137, 11760.

When checking ref. 43, no PVDF-TrFE random copolymers and P2VP-b-(PVDF-TrFE)-b-P2VP triblock copolymer are reported. Also, the molecular weight for the PS-b-(PVDF-TrFE)-b-PS triblock copolymer does not match that in ref. 43. So, what are molecular weights of P2VP and PS blocks? How is PVDF-TrFE molecular weight determined? If PS standards are used for SEC, the molecular weight is not absolute molecular weights. However, if PS and P2VP standards are used, and the PS and P2VP block molecular weights are determined by SEC first, then the PVDF-TrFE molecular weight calculated from NMR shall be the absolute average molecular weight.

How are the VDF/TrFE composition determined? ^{19}F NMR? Data needs to be shown at least in the Supporting Information.

It is better to use P2VP-b-P(VDF-TrFE)-b-P2VP and PS-b-P(VDF-TrFE)-b-PS to represent the ABA triblock copolymers, rather than P(VDF-TrFE)-b-P2VP or PS. For different composition, one can use P(VDF_x-TrFE_y).

3. Morphology: If the PVDF-TrFE composition is about 60 vol% for both P2VP and PS triblock copolymers, it is difficult to understand why the PS triblock copolymer form a spherical structure, rather than a disordered bicontinuous structure. From the SAXS result at 25 °C (Fig. 3a), one can still see the lamellar structure, although the 2nd order peak is relatively weak. What is the SAXS profile for the PS triblock copolymer at 170 °C? This result should be given to prove that the morphology is a spherical one.

4. DSC result: To clearly tell the Curie temperature (T_c) for different PVDF-TrFE samples, one should use the heating process rather than the cooling process. This is because upon cooling, Curie temperatures are quite close for different VDF/TrFE compositions (see Furukawa, et al., Adv. Colloid Int. Sci. 1997, 71-2, 183). Only upon heating, the Curie transitions are clear and have distinct T_c values.

Page 7, line 181: "compared to the parent homopolymers". It should be neat PVDF-TrFE random copolymers, not homopolymers.

5. WAXD results in Fig. 4: The peak-fitting results for PVDF-TrFE are not accurate obviously. This can be told from different amorphous peak shapes for different polymers. Supposedly, P2VP and PS are amorphous and should show two amorphous peaks. One is around 7.1 nm^{-1} for inter-ring average distance and the other is around 13.7 nm^{-1} for inter-main chain average distance. However, the 7.1 nm^{-1} peaks are not shown. Also, for amorphous PVDF, the inter-main chain average distance shall be different from P2VP and PS (i.e., usually around 13 nm^{-1} in the solid state and 12 nm^{-1} in the molten state). These amorphous peaks shall be taken into account; however, they were not. Therefore, the crystallinity calculations are not accurate.

Instead of using WAXD for crystallinity calculation, one should use DSC for crystallinity calculation. For PVDF-TrFE 65/35, the heat of fusion for 100% perfect crystals is around 42 J/g (see: Macromolecules 2003, 36, 7220). In this way, the crystallinity shall be more accurate.

Fig. 4d: The authors explain the double peaks for PVDF-TrFE 50/50 as a mixed phase (i.e., paraelectric + cooled phase). However, this is not true. It can be explained as the single cooled phase; see: Tashiro, K. Crystal structure and phase transition of PVDF and related copolymers. In Ferroelectric Polymers: Chemistry, Physics, and Applications, 1st ed.; Nalwa, H. S., Ed.; Marcel Dekker: New York, 1995; pp 63-182. Obviously, PVDF-TrFE 70/30 does easily develop the cooled phase, but favors the low-temperature ferroelectric phase.

6. Ferroelectric properties: The discussion using Pcomp, Epol, and Edepol is too complicated. Actually, the issue shall be relatively simple. P2VP has a high dielectric constant. Therefore, the electric field in PVDF-TrFE is higher in the P2VP triblock copolymer than in the PS triblock copolymer. As a result, PS forms a very good electric shield for PVDF-TrFE. That is why PS triblock copolymer appears to be linear dielectric and P2VP triblock copolymer is ferroelectric.

When change to PVDF-TrFE 50/50, the spontaneous polarization decreases as compared to PVDF-TrFE 70/30. As a result, an intermediate behavior is observed, i.e., double loops. However, this behavior is not antiferroelectric like, but rather ferrielectric like, because the middle linear portion does not exist.

Fig. 5d and 5e should be swapped. Otherwise, they do not match with the main text discussion.

7. Supporting Information:

- a) Fig. S3: Temperature should be accurately shown on the y-axes for both WAXD and SAXS.
- b) Fig. S4: It should clearly state that this sample is after thermal annealing in the paraelectric phase at $120 \text{ }^\circ\text{C}$ for x amount of time. Also, why the maximum polarization decreased after annealing as compared to the as-crystallized samples (see Fig. 5c)?

Overall, I consider that this manuscript does not contain enough novelty, and some results are not accurate. Particularly, these triblock copolymers are not suitable for electric energy storage application at all. I would recommend rejection.

Reviewer #2 (Remarks to the Author):

The work presented by Terzic et al. reports about a new way to tune ferroelectric response of PVDF based polymers, by the by incorporating insulating polymer chains at the chain ends in a form of block copolymers. The chemical synthetic approach (CuAAC click chemistry for producing PVDF-based block copolymers) has been previously presented by the same authors elsewhere (see ref 43); here this approach is applied to the specific problem of producing polymer with desired ferroelectric properties, elucidating in details chemical-physical aspects and mechanisms. For this reason I think the work contain sufficient novelty to be published on Nature Communications. Form a scientific point of view the paper is solid and cannot find any major shortcoming. The paper is also well written and easy to follow; reference are more than adequate.

I have just a few minor suggestions to improve the manuscript as follow:

- Figure 5 is wrongly referenced in the text: in particular all the item are shifted of a position (i.e. Fig 5.a should be Fig 5.b, Fig 5.b should be Fig 5.c .. and so on). Probably the Fig 5.a was added in a later stage of the manuscript writing, without updating the text.
- In Figure 4 it should be beneficial add on top o the graph the name of the relative polymer (as in Fig 5), for more easy reading of the figure.
- I see that the measurements on pristine P(VDF-TrFE) 50:50 is not discussed in the text while its copolymer with P2VP yes. The only point in which P(VDF-TrFE) 50:50 appears is in Fig 3.g. I think it is important include the characterization of P(VDF-TrFE) 50:50, at least as reference. Some of the info could be also simply added to SI if not too relevant to the discussion.

Reviewer #3 (Remarks to the Author):

Manuscript (MS) NCOMMS-18-27193 entitled "Novel electroactive materials with tunable response based on block copolymer self-assembly" reports a comprehensive study on the structure-relationship properties of original block copolymers based on poly(VDF-co-TrFE) sequence and PS and P2VP and the different behaviors, ferroelectric, antiferroelectric-like and linear dielectric, readily achieved by simple adjustment of the polarity of both blocks, showing the vast potential of such block copolymers for numerous applications in field of piezo-, ferro, or antiferroelectrical devices. Both the presences of PS and P2VP onto the electric properties are compared and discussed. The incorporation of P2VP chains at both ends of poly(VDF-co-TrFE) central block and their phase separation do not induce drastic changes in the shape of the D-E loops. An exchange of P2VP with PS of the same molecular weight generates considerably different switching characteristics. D-E loops become narrower, resembling linear dielectric behavior, with almost zero remanent polarization and the maximum polarization lower than for P2VP-b-Poly(VDF-co-TrFE) 70:30.

The corresponding author is an expert on the synthesis of fluorinated block copolymers (see the excellent review: *J Polym Sci Part A Polym Chem* 2014, 52, 2861–77) and on the assessment of their electroactive properties.

This MS combines various fields of research ranging from the macromolecular chemistry (synthesis and characterization of original block copolymers) and physical (electroactive) and thermal properties of the resulting block copolymers. Thus, it can attract a wide range of readers and is useful for Nature Communications. The main goal is to taylor the ferroelectric response of poly(VDF-co-TrFE) copolymers based on covalently linked of functional insulating chains to the chain ends instead of the backbone to get A-B-A triblock copolymers.

The Science disclosed is of high level and much care has been taken on the presentation of figures that clearly support the features and findings of this MS. The reviewer appreciates Figure 1 that is quite relevant and summarizes the goal of the MS.

For these reasons, I recommend that MS suitable for publication in Nature Comm. and minor revisions are required, as follows:

1. Abstract should be rewritten since the first four sentences are better suited for the introduction and thus should be shifted there. It would be better to included key features reported by the MS. The conclusion section is fine, concise and informative. Heading "Introduction" at the beginning of

the introduction is missing.

2. This work is well compared to previous ones with well cited references. However, the authors should cite WO 2013160621A1 patent that discloses a poly(VDF-co-TrFE) containing block copolymers by controlled radical (co)polymerization (or RDRP) of trifluoroethylene. They should also include the following references: *Nanoscale*, 2013, 5, 184–92; *Macromolecules*, 2015, 48, 7861–787; *Macromolecules*, 2017, 50, 503-511, *Polym. Chem.* 2017, 8, 1017-1027, *Polym. Chem.* 2017, 8, 1017-1025, and *Macromolecules*, 2017, 50, 3313–3322.
3. Keeping ester group from BPO is questionable and not well explained (as well is previous articles from the same authors). A simple IR spectrum could avoid that doubt. In addition, the spectra on the azido derivatives should evidence the presence of its frequency.
4. Though is a small amount, care should be take on use of PDMETA ligand that may be basic to induce some dehydrofluorination of VDF units.
5. ¹⁹F NMR spectra should be supplied in SI and should indicate the calculations of VDF and TrFE contents. I donot think that those of block copolymers are necessary but they can alos be inserted.
6. SEC should be recoded in DMF (it is not sure that poly(VDF-co-TrFE) is totally soluble in THF).
7. In page 228, it would be good to supply the equation to calculate “non-crystalline PS and P2VP” block.
8. Which was the solvent used in the first step? Any transfer? Please comment.
9. In the experimental part, the colors and states (I assume all powders) should be mentioned.
10. I am surprised by the low dispersity of poly(VDF-co-TrFE) copolymers since they are prepared by conventional radical copolymerization.

We would like to thank the Reviewers for the careful and thorough reading of this manuscript and their valuable comments as well as constructive suggestions. We have addressed all issues indicated in the review report below and believe that our revised manuscript meets the journal publication requirements.

Answers to the Reviewer #1

1. The objective of this work is ill-defined. In the introduction, it states that PVDF (as well as graft copolymers) is not suitable for uniform dispersion of nanoparticles in polymers (i.e., nanocomposites). This is not true! Actually, PVDF is polar and hydrophobic at the same time, and can be much better than simple hydrophobic polymers such as PS or even PMMA for polymer nanocomposites.

Answer: We strongly disagree with the statement that the objective of our work is ill-defined. The statement of the Reviewer that PVDF is a good polymer for the dispersion of nano-objects and the preparation of nanocomposites (for different applications including dielectric nanocomposites, multiferroic, piezoelectric) is wrong and he/she is also not supporting this statement by any published work. In contrast to his/her opinion, in the past multiple research projects were conducted in order to improve the dispersibility of nano-objects in PVDF. In Chem. Mater., 2010, 22 (18), pp 5350–5357, J. Mater. Chem. A, 2014,2, 5244-5251, Adv. Funct. Mater. 2015, 25, 3505–3513 and multiple other papers it is clearly stated that “The unique molecular structures of fluorinated polymers consisting of tightly packed fluorine atoms give rise to low surface energy and interchain forces. As a result, fluorinated polymers generally demix with most organic and inorganic materials, and the dispersion of dopants in the fluorinate polymer matrix always becomes problematic. That is the reason why various research groups report new methods to improve the dispersibility of nanoparticles. If we consider the molecular structure of P2VP that already demonstrated its ability for good and selective dispersion of nanoparticles (Macromolecules 2016, 49, 3352-3360.), it is clear that block copolymers offer something that other materials do not, including selective dispersion of nano-objects, while additionally having high potential for the alignment (electric, magnetic field, shear alignment) and the anisotropy of properties. The reviewer is additionally being contradictory in his/her comments since he/she in the second question uses the following reference JACS 2015, 137, 11760., in which it is clearly stated that strong contact of nanoparticles with nanofiller is of great interest, and it is difficult to achieve.

Regardless of this inaccurate or incorrect statement, this work does not focus on PVDF-based nanocomposites at all. This should be irrelevant.

Answer: Indeed, the work does not focus on the PVDF-based nanocomposites. However, PVDF nanocomposites are one of the mostly used PVDF-based materials and their application depends on the type of the nanofillers. Therefore, one of the main drawbacks of PVDF to obtain high quality dispersion of nanocomposites should be addressed in the Manuscript. We again removed the statement about nanocomposites in the second paragraph and made it more general, since we did not show the beneficial property of block copolymers to selectively disperse nano-objects.

Next, the Introduction discusses about PVDF-graft copolymers. However, the authors did not fully understand the purpose of those PVDF-graft copolymers. They are intended for electric energy storage (e.g., capacitors), which is the application of linear dielectrics, not ferroelectrics (which is nonlinear dielectrics). Actually, ferroelectricity is UNDESIREED for capacitors, because most of the charged energy cannot be discharged due to energy storage in aligned/poled ferroelectric domains (also see the Discussion section). It is true that PVDF block copolymers are different from PVDF graft copolymers, because the ferroelectric properties can be largely preserved for the PVDF blocks. However, they are simply not useful at all for capacitors!

Answer: The authors again have to disagree with the statement of the reviewer about not understanding the current purpose of graft copolymers. However, as we as well mentioned in the text, the ultimate goal is to have materials that can have all types of electrical behavior that can be changed by simple adjustment. Indeed, using graft copolymers, as the Reviewer agrees, it is not possible to preserve ferroelectricity, not even at really small contents of insulating polymer due to reasons mentioned in the manuscript. On contrary, block copolymers, as we also showed in the Manuscript, can show all possible behaviors. Additionally, the Reviewer states that block copolymers cannot be used for energy storage without stating the reason why. We again have to disagree, since the block copolymers with PS show exactly linear behavior with almost zero remanent polarization. Additionally, block copolymer with P2VP and P(VDF-TrFE) 50:50 shows double hysteresis loop that is also recognized as beneficial behavior for energy storage since in this way slimmer hysteresis loops are obtained.

Third, if these PVDF-TrFE block copolymers are intended for piezoelectric applications, they are not good either, because the dipole density is reduced by the nonferroelectric blocks.

Answer: Even though we did not mention piezoelectric applications in the text, the material can be used for energy harvesting. We believe that block copolymers after alignment using shear forces, or substrate pretreatment can be used as self-poled nanogenerator, since it is well known that after crystallization confinement in nanodomains crystals have preferential orientation to the crystalline amorphous boundary and therefore can align the dipoles without applying long poling pretreatment, which is quite an improvement. It is also mentioned in the text that with any other approach like graft copolymer or blend, inclusion of this high amount of insulating polymer results in complete disruption of crystallization and loss of the ferroelectric properties. Here, with purpose, we went to one extreme, of course that for instance smaller amounts of P2VP can give higher polarization, while keeping functionality. Additionally, the possibility of materials to be used as piezoelectric generator does not depend only on the concentration of dipoles but also on the surface effects. In Science Advances DOI: 10.1126/sciadv.1602902 even though insulating graft chains were incorporated, the polymer demonstrated better energy harvesting capability.

Finally, it is not clear what these PVDF-TrFE block copolymers are good for? What is the exact goal of this work?

Answer: We mentioned in the conclusion "The incorporation of the functional insulating block does not only grant the tunable response of the ferroelectric polymer, but can potentially deliver additional benefits to the material, such as improved dispersion of nano-objects, or any other functional component using supramolecular approaches, better adhesion to the

electrodes, reduced conducting and dielectric losses and better film formation.” Additionally, as it can be seen from Figure 5 all block copolymers demonstrated higher breakdown strength compared to copolymer of even higher molecular weight. At the end we also discussed that the polymers we used are just a proof of concept. Having in mind large variety of functional block that can add a new property to the material, we think that our contribution presents a good guideline on what should be taken in consideration while selecting second functional block. That is one of the reasons why we spent quite some part of the Manuscript on understanding the physical phenomena behind this change in behavior after incorporation of an insulating block.

We now hope that it is clear why we think block copolymer are good materials for various applications and we feel sorry that the Reviewer does not share our opinion.

2. Synthesis: The click chemistry to make PVDF block copolymers is not very new. A similar (although not exactly the same) approach has already been reported; see: JACS 2015, 137, 11760.

Answer: Indeed, click chemistry is not for the first time used to synthesize PVDF based copolymers, as Reviewer #2 noted as well. We as well did not claim in this paper that we used click chemistry for the first time, but that we developed a versatile method for the synthesis of PVDF based block copolymers that can also work at higher molecular weights. The method in the paper the Reviewer is mentioning is indeed a click reaction and we will include it in the references. However, this method is used for preparation of PVDF based star copolymers rather than block copolymers. Additionally, the molecular weights obtained in the paper are drastically lower than we obtained. We have to emphasize that this is the first time that PVDF-based block copolymers are prepared with satisfying phase separation and possible film formation, that with polymers of only 3000 g mol^{-1} used in the paper Reviewer used for argumentation is not possible. Additionally, this is, to our knowledge so far, the only method that can give P(VDF-TrFE) based block copolymers, since long induction periods of ITP and RAFT polymerization will cause significant compositional drift.

When checking ref. 43, no PVDF-TrFE random copolymers and P2VP-b-(PVDF-TrFE)-b-P2VP triblock copolymer are reported. Also, the molecular weight for the PS-b-(PVDF-TrFE)-b-PS triblock copolymer does not match that in ref. 43. So, what are molecular weights of P2VP and PS blocks? How is PVDF-TrFE molecular weight determined? If PS standards are used for SEC, the molecular weight is not absolute molecular weights. However, if PS and P2VP standards are used, and the PS and P2VP block molecular weights are determined by SEC first, then the PVDF-TrFE molecular weight calculated from NMR shall be the absolute average molecular weight.

Answer: We included the synthesis procedure in the renewed version of the Experimental part, together with the calculated molecular weights of pristine P2VP and PS. The molecular weight of P(VDF-TrFE) is calculated from GPC values which is the standard method for determining the molecular weight of polymers, nowadays. It is a well-known fact that if the polymer has molecular weights as high as we report, the use of ^1H NMR for end group analysis would not be precise (theoretically NMR should be used till $10\,000 \text{ g mol}^{-1}$). The molecular weight of the block copolymers is calculated from the GPC values of P(VDF-TrFE) and using the weight fractions of P(VDF-TrFE) calculated from ^1H NMR, which is a common method for the determination of the molecular weight of block copolymers. Just using GPC is not accurate bearing in mind that

P(VDF-TrFE) gives negative, while other polymers positive, RI signal. We clarified this in the Table 1.

How are the VDF/TrFE composition determined? ¹⁹F NMR? Data needs to be shown at least in the Supporting Information.

Answer: VDF/TrFE compositions are calculated from ¹H NMR, but as the Reviewer suggests the composition can be determined from ¹⁹F NMR, as well. We included the method we used for calculation inside Supporting Info.

It is better to use P2VP-b-P(VDF-TrFE)-b-P2VP and PS-b-P(VDF-TrFE)-b-PS to represent the ABA triblock copolymers, rather than P(VDF-TrFE)-b-P2VP or PS. For different composition, one can use P(VDFx-TrFEy).

Answer: We understand the comment of Reviewer that the suggested notation is more accurate and less confusing for the readers. We made changes in the text accordingly.

3. Morphology: If the PVDF-TrFE composition is about 60 vol% for both P2VP and PS triblock copolymers, it is difficult to understand why the PS triblock copolymer form a spherical structure, rather than a disordered bicontinuous structure. From the SAXS result at 25 °C (Fig. 3a), one can still see the lamellar structure, although the 2nd order peak is relatively weak. What is the SAXS profile for the PS triblock copolymer at 170 °C? This result should be given to prove that the morphology is a spherical one.

Answer: We understand the point of Reviewer about for us as well unexpected formation of the spherical phase. The same morphology is observed in previous paper of ours (*Polym. Chem.* (2018). doi:10.1039/C8PY00742J) where we also show the SAXS profile of block copolymer in the melt. We can only speculate about the reason for the formation of such a morphology. We believe that the high polydispersity of middle block together with slight amount of diblock copolymers (already discussed in previous paper) can stabilize the structure with the highest curvature as previously reported by many authors (*J. Phys. Chem. B*, 2012, 116 (40), pp 12357–12371, *J. Am. Chem. Soc.* 2012, 134, 3834.) . Low segregation strength and higher polydispersity can induce that the formed structure varies from the expected morphology. Indeed, the formation of a lamellar phase in melt by the PS block copolymer would have saved us a few months of work in trying to find a way to obtain the same morphology. Additionally, the theoretical and experimental results on the crystallization of block copolymers show that the crystallization inside lamellar phase results in heterogeneous crystallization and no undercooling as we observe in P2VP lamellar block copolymers. On contrary, crystallization inside spherical morphology results in homogeneous nucleation and strong undercooling, as it is observed for PS based block copolymer. Therefore, we strongly believe that morphology we obtained for PS block copolymer is not lamellar as it is also proven by TEM.

4. DSC result: To clearly tell the Curie temperature (T_c) for different PVDF-TrFE samples, one should use the heating process rather than the cooling process. This is because upon cooling, Curie temperatures are quite close for different VDF/TrFE compositions (see Furukawa, et al.,

Adv. Colloid Int. Sci. 1997, 71-2, 183). Only upon heating, the Curie transitions are clear and have distinct T_c values.

Answer: We understand the comment of Reviewer, and we assume that Reviewer is worried about the existence of two Curie transitions upon cooling. However, we have to disagree with the comment that the Curie transition can be better observed upon heating. Actually, the Curie transition for P(VDF-TrFE) we synthesized with 20 wt. % of TrFE upon heating overlaps with melting peak. Checking the, temperature resolved WAXS showed no full transition before melting, while upon cooling the Curie transition is clearly visible. Additionally, for P(VDF-TrFE) 50:50, no Curie transition was observed in the heating scan as well. Of course, we did not name the peak on DSC T_c before first checking the WAXS profiles. We also think that for the purpose of proving the spherical morphology in block copolymer with PS and to see the mechanism of crystallization it is necessary to use the cooling scans.

Page 7, line 181: "compared to the parent homopolymers". It should be neat PVDF-TrFE random copolymers, not homopolymers.

Answer: We agree with Reviewer that P(VDF-TrFE) is a copolymer, and apologize for the mistake we made.

5. WAXD results in Fig. 4: The peak-fitting results for PVDF-TrFE are not accurate obviously. This can be told from different amorphous peak shapes for different polymers. Supposedly, P2VP and PS are amorphous and should show two amorphous peaks. One is around 7.1 nm^{-1} for inter-ring average distance and the other is around 13.7 nm^{-1} for inter-main chain average distance. However, the 7.1 nm^{-1} peaks are not shown. Also, for amorphous PVDF, the inter-main chain average distance shall be different from P2VP and PS (i.e., usually around 13 nm^{-1} in the solid state and 12 nm^{-1} in the molten state). These amorphous peaks shall be taken into account; however, they were not. Therefore, the crystallinity calculations are not accurate.

Answer: We agree with Reviewer that the amorphous part of block copolymers consists of both amorphous P(VDF-TrFE) and PS or P2VP, and we used this knowledge while doing the deconvolution. However, we used a global amorphous peak for the deconvolution summing both components, and used this principle while calculating the degree of crystallinity. The same is done in many papers related to the mixture of PVDF and amorphous component (Macromolecules, 2011, 44 (7), pp 2190–2199, Adv. Funct. Mater. 2011, 21, 3176–3188) We agree that the full range is not taken in account. However, we think that stronger effect on the values of the overall crystallinity should include parts above 20 nm^{-1} as opposed to parts below 8 nm^{-1} . Having in mind that all polymers are deconvoluted in the same q-region, we think that the mistake we are making in comparing their degrees of crystallinity are minimal. We can additionally prove this by comparing the values of degree of crystallinity obtained from DSC. The results we obtained are 39.5 % for pristine P(VDF₇₀-TrFE₃₀), 38% for P2VP-*b*-P(VDF₇₀-TrFE₃₀)-*b*-P2VP and 32.5 % for PS-*b*-P(VDF₇₀-TrFE₃₀)-*b*-PS. Having in mind that the crystallization mechanism for polymer with 50:50 ratio between VDF and TrFE is not the same as for the previous polymers we cannot use DSC to calculate degree of crystallinity (Nature 562, pages96–100 (2018)).

Instead of using WAXD for crystallinity calculation, one should use DSC for crystallinity calculation. For PVDF-TrFE 65/35, the heat of fusion for 100% perfect crystals is around 42 J/g (see: Macromolecules 2003, 36, 7220). In this way, the crystallinity shall be more accurate.

Answer: The crystallinity can indeed be calculated using DSC as we already mentioned in the previous answer and we did perform these measurements. The results we obtained are quite close to the values obtained using WAXS and also show the same trend in values, not changing the discussion and conclusions.

Fig. 4d: The authors explain the double peaks for PVDF-TrFE 50/50 as a mixed phase (i.e., paraelectric + cooled phase). However, this is not true. It can be explained as the single cooled phase; see: Tashiro, K. Crystal structure and phase transition of PVDF and related copolymers. In Ferroelectric Polymers: Chemistry, Physics, and Applications, 1st ed.; Nalwa, H. S., Ed.; Marcel Dekker: New York, 1995; pp 63-182. Obviously, PVDF-TrFE 70/30 does easily develop the cooled phase, but favors the low-temperature ferroelectric phase.

Answer: We understand the concern of Reviewer about the crystalline structure of P2VP-*b*-P(VDF₅₀-TrFE₅₀)-*b*-P2VP since it is discussed thoroughly in literature, whether it is a mixture of two phases or just one phase. However, just recently published paper in Nature (Nature 562, pages96–100 (2018)) shines more light on this issue reporting two phases inside one in P(VDF-TrFE) at the morphotropic phase boundary which is exactly the ratio between monomers we took in consideration. Additionally, the switching results we obtained for this block copolymers are in correspondence with the existing of two crystalline phases.

6. Ferroelectric properties: The discussion using Pcomp, Epol, and Edepol is too complicated. Actually, the issue shall be relatively simple. P2VP has a high dielectric constant. Therefore, the electric field in PVDF-TrFE is higher in the P2VP triblock copolymer than in the PS triblock copolymer. As a result, PS forms a very good electric shield for PVDF-TrFE. That is why PS triblock copolymer appears to be linear dielectric and P2VP triblock copolymer is ferroelectric.

Answer: The Reviewer has a point for simplifying the behavior of block copolymer, and we were highly aware of it while writing this manuscript. We actually used the same argument to explain the only slight increase in coercive field of the block copolymer. However, we think that exact physical phenomenon that takes place in phase separated block copolymers should be addressed to explain the tunable behavior. Especially this can be important for the future application of block copolymers, to understand what to expect from inclusion of functional block, since there are plenty of potentially attractive blocks for functionalization of ferroelectric materials.

When change to PVDF-TrFE 50/50, the spontaneous polarization decreases as compared to PVDF-TrFE 70/30. As a result, an intermediate behavior is observed, i.e., double loops. However, this behavior is not antiferroelectric like, but rather ferroelectric like, because the middle linear portion does not exist.

Answer: First part of the question is already discussed in previous Answer. We agree that the double loop behavior lack of the middle linear part in double loop behavior. However, the same nomenclature is used in the case of graft copolymers that show exactly the same shape. Additionally, in a just published paper Nature 562, pages96–100 (2018) the same shape of the hysteresis loop is called antiferroelectric like, or relaxor ferroelectric, we stick to one of them through the whole Manuscript.

Fig. 5d and 5e should be swapped. Otherwise, they do not match with the main text discussion.

Answer: We made a mistake by putting the Figure 5a in the last phase of manuscript preparation which caused confusion in Figure 5 notation. We thank the Reviewer and made changes in the text.

7. Supporting Information:

a) *Fig. S3: Temperature should be accurately shown on the y-axes for both WAXD and SAXS.*

Answer: We made changes with the temperature axis in the revised version of Manuscript.

b) *Fig. S4: It should clearly state that this sample is after thermal annealing in the paraelectric phase at 120 °C for x amount of time. Also, why the maximum polarization decreased after annealing as compared to the as-crystallized samples (see Fig. 5c)?*

Answer: The sample showed in Figure S4 is prepared after crystallization from the melt while the sample in Figure 5b is obtained as it is written by recrystallization in paraelectric phase for 30 minutes as it is stated in the text of the manuscript. However, for clarity reasons, we will incorporate it in the text of the Figure caption. The maximum polarization is lower for the block copolymer annealed in the melt compared for annealing at 120 degrees for few reasons. First, the increased polarization can be the consequence of the higher conductivity of the sample. Since annealing in the melt is removing defects in the structure of the solvent casted films better than after annealing at 120 degrees. Consequently, the sample annealed in the melt has lower leakage, as it is observed from the shape of the loops. Secondly, the crystallization at 120 degrees and from melt are giving even though both lamellar morphology but with different level of ordering and overall crystal orientation. It is also well known that annealing temperature influence dielectric constant of P(VDF-TrFE) and therefore can induce different maximum polarization.

Overall, I consider that this manuscript does not contain enough novelty, and some results are not accurate. Particularly, these triblock copolymers are not suitable for electric energy storage application at all. I would recommend rejection.

Answer: We are sorry that Reviewer does not find our manuscript novel enough. We believe that the block copolymer with PS has way higher polarization than current state-of-the-art biaxially oriented polypropylene while still having quite small losses-similar to graft copolymer and multilayered structures. It is interesting to mention that multilayered PVDF-PC structures are already in the process of the commercialization. We also wrote in the conclusion that our

work is just a proof of concept, since for ultimate practical application block copolymers of higher molecular weight should be prepared, but the results we obtained are highly encouraging not only for us but for the whole field of research and we hope it will inspire other researchers not only to apply our concept for material preparation but as well to discover new synthetical procedures to obtain materials with ultimate properties.

Answers to the Reviewer #2

Thanks for the nice words about our submitted manuscript and suggestions on how to improve it. We tried to make changes based on your remarks.

1. *Figure 5 is wrongly referenced in the text: in particular all the item are shifted of a position (i.e. Fig 5.a should be Fig 5.b, Fig 5.b should be Fig 5.c .. and so on). Probably the Fig 5.a was added in a later stage of the manuscript writing, without updating the text.*

Answer: We understand the reviewer's point. Indeed, Figure 5a was added in the last stage of manuscript writing without updating the text. We changed that in the new version of Manuscript.

2. *In Figure 4 it should be beneficial add on top of the graph the name of the relative polymer (as in Fig 5), for more easy reading of the figure.*

Answer: We agree with the Reviewer that additional name on the graph is making graphs clearer. We made changes in the new version of Manuscript.

3. *I see that the measurements on pristine P(VDF-TrFE) 50:50 is not discussed in the text while its copolymer with P2VP yes. The only point in which P(VDF-TrFE) 50:50 appears is in Fig 3.g. I think it is important include the characterization of P(VDF-TrFE) 50:50, at least as reference. Some of the info could be also simply added to SI if not too relevant to the discussion.*

Answer: We understand the concern of Reviewer and we included some of the data in the Supporting info and some as a reference.

Answer to the Reviewer #3

Thanks for nice comments about our manuscript, thorough analysis and comments. Additionally, we are happy that the objective of our manuscript is well understood from the Reviewer's side.

1. *Abstract should be rewritten since the first four sentences are better suited for the introduction and thus should be shifted there. It would be better to included key features reported by the MS.*

The conclusion section is fine, concise and informative. Heading "Introduction" at the beginning of the introduction is missing.

Answer: We understand Reviewer's point and we performed changes in revised Manuscript.

2. *This work is well compared to previous ones with well cited references. However, the authors should cite WO 2013160621A1 patent that discloses a poly(VDF-co-TrFE) containing block copolymers by controlled radical (co)polymerization (or RDRP) of trifluoroethylene. They should also include the following references: Nanoscale, 2013, 5, 184–92; Macromolecules, 2015, 48, 7861–787, Macromolecules, 2017, 50, 503-511, Polym. Chem. 2017, 8, 1017-1027, and Macromolecules, 2017, 50, 3313–3322.*

Answer: We agree with the Reviewer that not maybe all significant references are included in the Manuscript. We made changes in the revised version.

3. *Keeping ester group from BPO is questionable and not well explained (as well is previous articles from the same authors). A simple IR spectrum could avoid that doubt. In addition, the spectra on the azido derivatives should evidence the presence of its frequency.*

Answer: We understand Reviewer's point about the preservation of ester groups during the click reaction. In this paper P(VDF-TrFE) polymers used for the preparation of block copolymers have high molecular weight, and therefore it is difficult to determine end groups using IR. Anyway, to answer the question of the Reviewer, we performed IR measurement of block copolymers with significantly lower molecular weight than used in the Manuscript. It can be seen that the peak corresponding to the C=O vibration stays preserved after the click reaction. The vibration of C-O bond is simply not visible using FTIR since it overlaps with the strong peaks of polymer backbone. Additionally, in our previous paper *Polym. Chem.* (2018). doi:10.1039/C8PY00742J in Figure 2, we clearly show preservation of the peak at 4.68 ppm that corresponds to the VDF protons bonded to the ester group after click reaction. We also provide this Figure here. Therefore, we strongly believe that that this is enough proof that possible removal of the BPO ester group during the click reaction is not an issue for the preparation of block copolymers. The preservation of the end group can be also concluded from the GPC values, since cleavage of the end groups will result in presence of homopolymers and consequently a negative peak of the RI signal for block copolymers. The existence of azide end group is already proved in the previous publication of ours where we demonstrate the synthetic method (*Polym. Chem.* **2014**, *5*, 2219) so there is no doubt that we obtained azide terminated polymer.

Figure: $^1\text{H-NMR}$ spectra in acetone- d_6 showing the successful preparation of PtBA- b -PVDF- b -PtBA block copolymers starting from azide terminated PVDF. After click reaction the peak at 4.68 ppm corresponding to the protons of VDF bonded to the ester group does not change showing no removal of the end groups during click reaction.

4. Though is a small amount, care should be taken on use of PDMETA ligand that may be basic to induce some dehydrofluorination of VDF units.

Answer: We understand the concern of Reviewer about the possibility of PMDETA to cause dehydrofluorination, that can affect the number of dipoles and the electric response of the

polymers. Indeed, the dehydrofluorination is unavoidable in this procedure, which indeed gave an off-white product. However, it has been observed before that with as little as 0.1 % of dehydrofluorination, PVDF turns into a deep black polymer, as a result of the formation of conjugated C=C bonds within the polymer backbone. We tried to keep the amount of PMDETA during reaction as lower as possible while obtaining full conversion of the end groups. Therefore, only a slight change in color of our product occurred (light brown) indicating insignificant dehydrofluorination.

5. *¹⁹F NMR spectra should be supplied in SI and should indicate the calculations of VDF and TrFE contents. I do not think that those of block copolymers are necessary but they can also be inserted.*

Answer: We included ¹⁹F NMR spectra of pristine P(VDF-TrFE) copolymers and explained the calculation of the molar content of TrFE using ¹H NMR. The same can be done using, as the Reviewer indicated, using ¹⁹F-NMR. Both methods, as expected, gave the same result.

6. *SEC should be recoded in DMF (it is not sure that poly(VDF-co-TrFE) is totally soluble in THF).*

Answer: We understand the comment of the Reviewer related to the choice of solvent for GPC analysis, having in mind that PVDF homopolymer does not dissolve well in THF. However, the presence of TrFE units in the copolymer makes it better soluble in various solvents, including THF. Nowadays, THF is even used as a solvent of choice for the preparation of thin films containing P(VDF-TrFE) (Functional Materials. 21, 10, p. 1887-1894 8 p and Nature Materials volume 7, pages 547–550 (2008)). Having in mind that the molecular weight of our pristine P(VDF-TrFE) copolymers is at least 3 times lower than the commercial polymers, straightforward solubility is obtained, without using heating or extensive stirring. The reason we used THF specifically is that that solvent showed better separation of P2VP containing block copolymers compared to DMF, which demonstrated some aggregation on the column. As it is showed in Figure 1. we wanted to prove block copolymer formation by GPC and therefore we needed results in the same solvent, which was THF.

7. *In page 28, it would be good to supply the equation to calculate “non-crystalline PS and P2VP” block.*

Answer: We understand the comment of Reviewer. We just think it is maybe too crowded to put equation under Table 1. Therefore, we included it in the Supporting Information. In principle, the values of the integrals for aromatic peaks of PS and P2VP were compared with the TrFE peak, which corresponds to an already known number of protons.

8. *Which was the solvent used in the first step? Any transfer? Please comment.*

Answer: Based on the comments of Reviewer #1 we included the experimental procedure for the synthesis of P(VDF-TrFE) with chlorine end groups. The same backbiting reactions observed in our previous report for PVDF based block copolymers are observed in the case of TrFE. Any other transfer reactions are kept to a minimum since we used acetonitrile as solvent that doesn't contain proton that can be consumed by radicals and should not show transfer reactions. They are as well not observed in ^1H NMR. We tried as well to perform the reaction in dimethylcarbonate, which is by few authors highlighted as a solvent of choice for the synthesis of PVDF based polymers. However, strong peaks corresponding to the end groups made by chain transfer to this solvent are observed, so we decided to use acetonitrile.

9. *In the experimental part, the colors and states (I assume all powders) should be mentioned.*

Answer: We understand the comments of Reviewer and we made changes in the Experimental part.

10. *I am surprised by the low dispersity of poly(VDF-co-TrFE) copolymers since they are prepared by conventional radical copolymerization.*

Answer: We understand the reviewer's point about the low PDI values of the polymer made by free radical polymerization. This is dependent on a few factors. First, theoretically, the absence of the disproportionation reaction reduces the PDI by 0.5 and with only recombination and without chain transfer the value of PDI should be 1.5. It is well known fact that disproportionation as a termination mechanism does not occur for fluorinated monomers as VDF and TrFE. However, we cannot avoid the chain transfer reactions, as already mentioned in the text. Secondly, usage of the acetonitrile as solvent reduces the chain transfer reactions during polymerization, and therefore lower PDI can be achieved. Third, it is written in the new version of the Experimental part that methanol and dichloromethane are used for washing of the polymers. It is already proved that methanol dissolves lower molecular weight of P(VDF-TrFE) which is again leading to the reduction of PDI.

Reviewers' comments:

Reviewer #1 (Remarks to the Author):

Comments on Responses:

Reviewer 1

Response 1:

i/ii) Nanocomposites: The authors are not 100% correct about PVDF nanocomposites. The dispersion of nanofillers actually depends on the interaction with PVDF. For example, if the nanofillers are polar, the strong interaction with PVDF favors particle dispersion. For example, it is well-known that PVDF can strongly interact with organoclay, carbon nanotubes, and inorganic salts. These composites have relatively good dispersion of nanoparticles. If the nanofillers are less polar or nonpolar, poor dispersion will result. Either modification of nanofillers or PVDF should be pursued. Anyway, this topic obviously is outside the scope of this work. The reviewer welcomes the authors to remove the irrelevant statements (goals). Actually, these irrelevant statements will weaken the manuscript.

iii) Electric Energy Storage: The reviewer concerns about the authors setting the goal for this manuscript as electric energy storage (i.e., capacitors). The normal ferroelectric behavior in Fig. 5c is absolutely not suitable for electric energy storage. The double loop behavior in Fig. 5e is also not suitable, because the hysteresis loop loss is over 30-40%. The linear dielectric behavior in Fig. 5d should be desired for electric energy storage; however, the loop loss is still far greater than that for BOPP. For real capacitor application, it is generally required that the $\tan(\delta)$ should be lower than 0.003 (i.e., the value for Mylar films) and the loop loss should be less than 5% at 300 MV/m. Obviously, even the sample in Fig. 5d would not be able to work because the loop loss is estimated to be around at least 10% at 300 MV/m.

Therefore, the reviewer considers that different ferroelectric behaviors can be achieved for P2VP-PVDF-P2VP triblock copolymers. The authors should not use electric energy storage as the goal of this manuscript!

iv) Piezoelectric Property: When the reviewer looks into the Science Advances paper (DOI: 10.1126/sciadv.1602902), it should be incorrect that the dielectric constant increases with grafting more PtBA onto PVDF. This is again all other reported PVDF-graft copolymer results quoted in the manuscript. Basically, adding a low dielectric constant component, the dielectric constant of PVDF will decrease! The result of increased dielectric constant with grafting PtBA on PVDF is against science!

Regardless, piezoelectric property is much more complicated than the authors think. More importantly, piezoelectricity is outside the scope of this work. The authors should not state that these block copolymers could have better piezoelectricity.

v) Goal of the Work: Again, the reviewer considers that the merit of this work should be the self-assembly and various ferroelectric (i.e., nonlinear dielectric) to linear dielectric behaviors can be realized in the block copolymer system. All nanocomposites, electric energy storage, and piezoelectricity should not be over sold here!

Response 2 - Synthesis

i) RAFT: The response is okay.

ii) Molecular Weights: The response is okay.

iii) VDF/TrFE Composition: The response is okay.

iv) Name of Triblocks: The response is okay.

Response 3 - Morphology

The response is okay.

Response 4 – DSC Results

The authors did not even look into the reference that the reviewer provided. Upon cooling, there is

not much different in Curie temperatures for different TrFE amount samples. Also, we have studied P(VDF-TrFE) 80/20. The Curie point is 130C and the melting temperature is 158C. So, there is a good separation between the Curie and melting temperatures. The authors should supply DSC heating curves to convince the reviewer and readers.

Response 5 – WAXD

It shall be okay to consider that the 50/50 sample contains a very disordered phase or mixed phases. The XRD reflections are too broad to be accurate. However, a simple calculation of crystallinity by DSC shall be made to compare to results of WAXD.

Response 6 – Ferroelectricity

The responses shall be okay.

Response 7 – SI

- a) The response is okay.
- b) The conduction might be the reason.

Overall, the reviewer consider that the authors argue quite much, rather than adequately address the comments. At last major revision is needed for this manuscript.

Reviewer #2 (Remarks to the Author):

For what concerns my previous comments, I am happy to see that all the issues have been correctly addressed.

For this reason I endorse the publication of the paper.

Reviewer #3 (Remarks to the Author):

The revision of manuscript (MS) entitled “Novel electroactive materials with tunable response based on block copolymer self-assembly” has been much improved from the point by point answers to reviewers’ comments.

All the points raised in the previous round have been satisfactorily addressed and even several additional points have been also able to strengthen that study that is solid and comprehensive, well-presented, and well supported by rigorous characterizations. In addition, graphs and charts are well done. The present MS deserves to be published in Nature Communications.

We would like to thank the Reviewers again for their valuable comments as well as constructive suggestions. We have addressed all issues indicated in the review report below and believe that our revised manuscript meets the journal publication requirements. The major changes are highlighted grey in the revised version.

Answers to the Reviewer #1

Response 1:

i/ii) Nanocomposites: The authors are not 100% correct about PVDF nanocomposites. The dispersion of nanofillers actually depends on the interaction with PVDF. For example, if the nanofillers are polar, the strong interaction with PVDF favors particle dispersion. For example, it is well-known that PVDF can strongly interact with organoclay, carbon nanotubes, and inorganic salts. These composites have relatively good dispersion of nanoparticles. If the nanofillers are less polar or nonpolar, poor dispersion will result. Either modification of nanofillers or PVDF should be pursued.

Answer: We agree that PVDF shows strong interaction with fillers such as nanoclays and carbon nanotubes that in fact induced the formation of ferroelectric phases. We also agree that if the nanofillers are less polar it is possible either to modify the surface or to modify PVDF, and this is the main reason why we think that modification of PVDF by making block copolymers can be a good way to improve dispersity of less polar nanoparticles. Additionally, the use of block copolymers for the dispersion of NOs offers various advantages compared to the dispersion of NOs in homopolymers. The self-assembly of block copolymers allows exact control over the nanocomposite morphology, local environment, polymer-particle interaction and alignment of NOs that can lead to the anisotropy in properties. Again, we agree that since we did not report any results related to the dispersion of NOs we should remove this part from the introduction. Therefore, we performed significant changes in the Introduction in order to strengthen our manuscript.

Anyway, this topic obviously is outside the scope of this work. The reviewer welcomes the authors to remove the irrelevant statements (goals). Acutally, these irrelevant statements will weaken the manuscript.

Answer: We agree with the reviewer that the dispersion of nano-objects inside PVDF is not in scope of this manuscript, and therefore we have made changes in the text (see Introduction)

iii) Electric Energy Storage: The reviewer concerns about the authors setting the goal for this manuscript as electric energy storage (i.e., capacitors). The normal ferroelectric behavior in Fig. 5c is absolutely not suitable for electric energy storage. The double loop behavior in Fig. 5e is also not suitable, because the hysteresis loop loss is over 30-40%. The linear dielectric behavior in Fig. 5d should be desired for electric energy storage; however, the loop loss is still far greater than that for BOPP. For real capacitor application, it is generally required that the $\tan(\delta)$ should be lower than 0.003 (i.e., the value for Mylar films) and the loop loss should be less than 5% at 300 MV/m. Obviously, even the sample in Fig. 5d would not be able to work because the loop loss is estimated to be around at least 10% at 300 MV/m.

Therefore, the reviewer considers that different ferroelectric behaviors can be achieved for P2VP-PVDF-P2VP triblock copolymers. The authors should not use electric energy storage as the goal of this manuscript!

Answer: We thank the reviewer for his explanation and we agree that the application as energy storage is at the moment not feasible with this material and we have made changes in the text by completely changing the introduction.

iv) Piezoelectric Property: When the reviewer looks into the Science Advances paper (DOI: 10.1126/sciadv.1602902), it should be incorrect that the dielectric constant increases with grafting more PtBA onto PVDF. This is again all other reported PVDF-graft copolymer results quoted in the manuscript. Basically, adding a low dielectric constant component, the dielectric constant of PVDF will decrease! The result of increased dielectric constant with grafting PtBA on PVDF is against science!

Regardless, piezoelectric property is much more complicated than the authors think. More importantly, piezoelectricity is outside the scope of this work. The authors should not state that these block copolymers could have better piezoelectricity.

Answer: We thank the reviewer for pointing out the mistakes made in the paper we used as a reference. We agree that piezoelectric is a complicated property and it is out of the scope of this manuscript and therefore we have made changes in the Introduction.

v) Goal of the Work: Again, the reviewer considers that the merit of this work should be the self-assembly and various ferroelectric (i.e., nonlinear dielectric) to linear dielectric behaviors can be realized in the block copolymer system. All nanocomposites, electric energy storage, and piezoelectricity should not be over sold here!

Answer: We agree with the reviewer that a different goal would improve the manuscript and therefore we have made all the suggested changes in the Introduction. We also want to thank the reviewer for his points about the strong sides of our manuscript that helped us in rewriting Introduction.

Response 2 - Synthesis

i) RAFT: The response is okay.

ii) Molecular Weights: The response is okay.

iii) VDF/TrFE Composition: The response is okay.

iv) Name of Triblocks: The response is okay.

Response 3 - Morphology

The response is okay.

Answer: We are glad that the reviewer found our answers satisfying and thank for the improvement of the manuscript.

Response 4 – DSC Results

The authors did not even look into the reference that the reviewer provided. Upon cooling, there is not much different in Curie temperatures for different TrFE amount samples. Also, we have studied P(VDF-TrFE) 80/20. The Curie point is 130C and the melting temperature is 158C. So, there is a good separation between the Curie and melting temperatures. The authors should supply DSC heating curves to convince the reviewer and readers.

Answer: We have to apologize to the reviewer if there is a misunderstanding coming from our answer. The samples we studied have a lower molecular weight and are produced by a direct copolymerization of VDF and TrFe. Therefore, the melting temperature of P(VDF-TrFE) is more than 10 degrees lower than the commercially available polymers. Additionally, the Curie transition occurs at higher temperatures, probably due to a different distribution of TrFE units after the direct

copolymerization and dichlorination of CTFE units. Therefore, upon heating the Curie transition is not that well visible as upon cooling, which we try to demonstrate in the following Figure. It becomes obvious that upon heating the difference between Curie and the melting temperature is not visible in the DSC thermogram, whereas upon cooling the difference is highly visible. Also, even for P(VDF-TrFE) 70/30 the peak of the Curie transition slightly overlaps on DSC with the melting peak and is less defined as the one upon cooling. This, together with the fact that the position of the crystallization peak can tell us a lot about the mechanism is the reason why we are showing cooling scans and not the heating ones.

Fig. 1 DSC and temperature-resolved WAXS profiles of P(VDF-TrFE) 80/20(a) cooled from the melt at $10\text{ }^{\circ}\text{C min}^{-1}$, (b) heated at $10\text{ }^{\circ}\text{C min}^{-1}$ and of P(VDF-TrFE) 70/30 (c) cooled from the melt at $10\text{ }^{\circ}\text{C min}^{-1}$, (d) heated at $10\text{ }^{\circ}\text{C min}^{-1}$.

Response 5 – WAXD

It shall be okay to consider that the 50/50 sample contains a very disordered phase or mixed phases. The XRD reflections are too broad to be accurate. However, a simple calculation of crystallinity by DSC shall be made to compare to results of WAXD.

Answer: We understand the reviewer’s concern about the accuracy of deconvolution of this specific sample as the two peaks are quite close to each other. However, when we take a look at the WAXS profile of the pure polymer P(VDF-TrFE) 50/50, it is clear that two peaks appear. The increased crystallinity with the increase of TrFE amount is expected and it is a well-known fact that the increase of TrFE amount increases the degree of crystallinity.

As the reviewer suggests we included the values obtained by DSC in the manuscript and in the table.

Response 6 – Ferroelectricity

The responses shall be okay.

Response 7 – SI

a) The response is okay.

b) The conduction might be the reason.

Overall, the reviewer consider that the authors argue quite much, rather than adequately address the comments. At last major revision is needed for this manuscript.

Answer: We again thank the reviewer for all the help in improving the quality of the manuscript and we hope that the major changes we added are sufficient for acceptance of our manuscript.

Reviewer #2 (Remarks to the Author):

For what concerns my previous comments, I am happy to see that all the issues have been correctly addressed.

For this reason I endorse the publication of the paper.

Reviewer #3 (Remarks to the Author):

The revision of manuscript (MS) entitled “Novel electroactive materials with tunable response based on block copolymer self-assembly” has been much improved from the point by point answers to reviewers’ comments.

All the points raised in the previous round have been satisfactorily addressed and even several additional points have been also able to strengthen that study that is solid and comprehensive, well-presented, and well supported by rigorous characterizations. In addition, graphs and charts are well done. The present MS deserves to be published in Nature Communications.

Answer: We again thank the reviewers #2 and #3 for their remarks and encouraging words about our manuscript.

REVIEWERS' COMMENTS:

Reviewer #1 (Remarks to the Author):

I consider that the manuscript is okay to be accepted.

Reviewer #3 (Remarks to the Author):

I have read the answers to reviewers and whatever the topics addressed, molecular weights, VDF/TrFE compositions, morphologies, DSC and WAXD results, nanocomposites, electric energy storage, goal of the work, I have found the right and pertinent answers which are well convincing and accurate. In addition, the changes made in the introduction sound well done (regarding question of comparison with graft copolymers on piezoelectrical properties).

The SI is useful and also guides the reviewer. The study is well supported by rigorous characterizations

Again, as I mentioned last month, synthesizing P2VP-b-PVDF-b-P2VP triblock copolymers is quite original and the properties sound suitable for new ferroelectric devices.

To my opinion, the re-revision of manuscript (MS) takes the reviewers' comments into account and the revised MS is much improved.

The present MS deserves to be published in Nature Communications.

Prof. Bruno Ameduri

Reviewer #1 (Remarks to the Author):

I consider that the manuscript is okay to be accepted.

Answer: We are glad that the Reviewer #1 is convinced and agrees that our Manuscript should be accepted in Nature Communications. We are as well grateful on his significant amount of comments that for sure improved quality of our Manuscript.

Reviewer #3 (Remarks to the Author):

I have read the answers to reviewers and whatever the topics addressed, molecular weights, VDF/TrFE compositions, morphologies, DSC and WAXD results, nanocomposites, electric energy storage, goal of the work, I have found the right and pertinent answers which are well convincing and accurate. In addition, the changes made in the introduction sound well done (regarding question of comparison with graft copolymers on piezoelectrical properties). The SI is useful and also guides the reviewer. The study is well supported by rigorous characterizations

Again, as I mentioned last month, synthesizing P2VP-b-PVDF-b-P2VP triblock copolymers is quite original and the properties sound suitable for new ferroelectric devices. To my opinion, the re-revision of manuscript (MS) takes the reviewers' comments into account and the revised MS is much improved. The present MS deserves to be published in Nature Communications.

Prof. Bruno Ameduri

Answer: We are glad that prof. Ameduri found our manuscript convincing and novel and for his recommendation of publishing our manuscript. We agree that after the addressing the Reviewers comments our Manuscript became highly improved.